# The Double-Edged Sword of Implicit Bias: Generalization vs. Robustness in ReLU Networks

**Spencer Frei**[*]
UC Davis
sfrei@ucdavis.edu

**Gal Vardi**[*]
TTI-Chicago and Hebrew University
galvardi@ttic.edu

**Peter L. Bartlett**
UC Berkeley and Google DeepMind
peter@berkeley.edu

**Nathan Srebro**
TTI-Chicago
nati@ttic.edu

## Abstract

In this work, we study the implications of the implicit bias of gradient flow on generalization and adversarial robustness in ReLU networks. We focus on a setting where the data consists of clusters and the correlations between cluster means are small, and show that in two-layer ReLU networks gradient flow is biased towards solutions that generalize well, but are vulnerable to adversarial examples. Our results hold even in cases where the network is highly overparameterized. Despite the potential for harmful overfitting in such settings, we prove that the implicit bias of gradient flow prevents it. However, the implicit bias also leads to non-robust solutions (susceptible to small adversarial $\ell_2$-perturbations), even though robust networks that fit the data exist.

## 1 Introduction

A central question in the theory of deep learning is how neural networks can generalize even when trained without any explicit regularization, and when there are more learnable parameters than training examples. In such optimization problems there are many solutions that label the training data correctly, and gradient descent seems to prefer solutions that generalize well [Zha+17]. Thus, it is believed that gradient descent induces an *implicit bias* towards solutions which enjoy favorable properties [Ney+17]. Characterizing this bias in various settings has been a subject of extensive research in recent years, but it is still not well understood when the implicit bias provably implies generalization in non-linear neural networks.

An additional intriguing phenomenon in deep learning is the abundance of *adversarial examples* in trained neural networks. In a seminal paper, Szegedy et al. [Sze+14] observed that deep networks are extremely vulnerable to adversarial examples, namely, very small perturbations to the inputs can significantly change the predictions. This phenomenon has attracted considerable interest, and various attacks (e.g., [GSS15; CW17; Pap+17; ACW18; CW18; Wu+20]) and defenses (e.g., [Pap+16; KGB17; Mad+18; WK18; CH20; WRK20]) were developed. However, the fundamental principles underlying the existence of adversarial examples are still unclear, and it is believed that for most tasks where trained neural networks suffer from a vulnerability to adversarial attacks, there should exist other neural networks which can be robust to such attacks. This is suggestive of the possible role of the optimization algorithms used to train neural networks in the existence of adversarial examples.

In this work, we study the implications of the implicit bias for generalization and robustness in ReLU networks, in a setting where the data consists of clusters (i.e., Gaussian mixture model) and the

---

[*]Equal contribution.

37th Conference on Neural Information Processing Systems (NeurIPS 2023).

correlations between cluster means are small. We show that in two-layer ReLU networks trained with the logistic loss or the exponential loss, gradient flow is biased towards solutions that generalize well, albeit they are non-robust. Our results are independent of the network width, and hence they hold even where the network has significantly more parameters than training examples. In such an overparameterized setting, one might expect harmful overfitting to occur, but we prove that the implicit bias of gradient flow prevents it. On the flip side, in our setting the distances between clusters are large, and thus one might hope that gradient flow will converge to a robust network. However, we show that the implicit bias leads to non-robust solutions.

Our results rely on known properties of the implicit bias in two-layer ReLU networks trained with the logistic or the exponential loss, which were shown by Lyu and Li [LL20] and Ji and Telgarsky [JT20]. They proved that if gradient flow in homogeneous models (which include two-layer ReLU networks) with such losses reaches a small training loss, then it converges (in direction) to a KKT point of the maximum-margin problem in parameter space. We show that in clustered data distributions, with high probability over the training dataset, every network that satisfies the KKT conditions of the maximum-margin problem generalizes well but is non-robust. Thus, instead of analyzing the trajectory of gradient flow directly in the complex setting of training two-layer ReLU networks, we demonstrate that investigating the KKT points is a powerful tool for understanding generalization and robustness. We emphasize that our results hold in the *rich* (i.e., *feature learning*) regime, namely, the neural network training does not lie in the kernel regime, and thus we provide guarantees which go beyond the analysis achieved using NTK-based results.

In a bit more detail, our main contributions are the following:

- Suppose that the data distribution consists of $k$ clusters, and the training dataset is of size $n \geq \tilde{\Omega}(k)$. We show that with high probability over the size-$n$ dataset, if gradient flow achieves training loss smaller than $\frac{1}{n}$ at some time $t_0$, then it converges in direction to a network that generalizes well (i.e., has a small test error). Thus, gradient-flow-trained networks cannot harmfully overfit even if the network is highly overparameterized. The sample complexity $\tilde{\Omega}(k)$ in this result is optimal (up to log factors), since we cannot expect to perform well on unseen data using a training dataset that does not include at least one example from each cluster.

- In the same setting as above, we prove that gradient flow converges in direction to a non-robust network, even though there exist robust networks that classify the data correctly. Specifically, we consider data distributions on $\mathbb{R}^d$ such that the distance between every pair of clusters is $\Omega(\sqrt{d})$, and we show that there exists a two-layer ReLU network where flipping the output sign of a test example requires w.h.p. an $\ell_2$-perturbation of size $\Omega(\sqrt{d})$, but gradient flow converges to a network where we can flip the output sign of a test example with an $\ell_2$-perturbation of size much smaller than $\sqrt{d}$. Moreover, the adversarial perturbation depends only on the data distribution, and not on the specific test example or trained neural network. Thus, the perturbation is both *universal* [Moo+17; Zha+21] and *transferable* [Liu+17; AM18]. We argue that clustered data distributions are a natural setting for analyzing the tendency of gradient methods to converge to non-robust solutions. Indeed, if positive and negative examples are not well-separated (i.e., the distances between points with opposite labels are small), then robust solutions do not exist. Thus, in order to understand the role of the optimization algorithm, we need a setting with sufficient separation between positive and negative examples.

The remainder of this paper is structured as follows: Below we discuss related work. In Section 2 we provide necessary notations and background, and introduce our setting and assumptions. In Sections 3 and 4 we state our main results on generalization and robustness (respectively), and provide some proof ideas, with all formal proofs deferred to the appendix. We conclude with a short discussion (Section 5).

**Related work**

**Implicit bias in neural networks.** The literature on implicit bias in neural networks has rapidly expanded in recent years, and cannot be reasonably surveyed here (see Vardi [Var22] for a survey). In what follows, we discuss results that apply to two-layer ReLU or leaky-ReLU networks trained with gradient flow in classification settings.

By Lyu and Li [LL20] and Ji and Telgarsky [JT20], homogeneous neural networks (and specifically two-layer ReLU networks, which are the focus of this paper) trained with exponentially-tailed classification losses converge in direction to a KKT point of the maximum-margin problem. Our analysis of the implicit bias relies on this result. We note that the aforementioned KKT point may not be a global optimum of the maximum-margin problem [VSS22]. Recently, Kunin et al. [Kun+22] extended this result by showing bias towards margin maximization in a broader family of networks called *quasi-homogeneous*.

Lyu et al. [Lyu+21], Sarussi, Brutzkus, and Globerson [SBG21], and Frei et al. [Fre+23b] studied implicit bias in two-layer leaky-ReLU networks with linearly-separable data, and proved that under some additional assumptions gradient flow converges to a linear classifier. Chizat and Bach [CB20] studied the dynamics of gradient flow on infinite-width homogeneous two-layer networks with exponentially-tailed losses, and showed bias towards margin maximization w.r.t. a certain function norm known as the variation norm. Phuong and Lampert [PL20] studied the implicit bias in two-layer ReLU networks trained on *orthogonally separable data*.

Safran, Vardi, and Lee [SVL22] proved implicit bias towards minimizing the number of linear regions in univariate two-layer ReLU networks, and used this result to obtain generalization bounds. Similarly to our work, they used the KKT conditions of the maximum-margin problem in parameter space to prove generalization in overparameterized networks. However, our setting is significantly different. Implications of the bias towards KKT points of the maximum-margin problem were also studied in Haim et al. [Hai+22], where they showed that this implicit bias can be used for reconstructing training data from trained ReLU networks.

**Theoretical explanations for non-robustness in neural networks.** Despite much research, the reasons for the abundance of adversarial examples in trained networks are still unclear [GSS15; FFF18; Sha+19; Sch+18; KH18; Bub+19; AL21; Wan+20; Sha+20; SMB21; Sin+21; Wan+22; DB22]. Below we discuss several prior theoretical works on this question.

In one line of work, it has been shown that small adversarial perturbations can be found for any fixed input in certain neural networks with random weights (drawn from the Gaussian distribution) [DS20; Bub+21; BBC21; MW22]. These works differ in the assumptions about the width and depth of the networks as well as the activation functions considered. However, since trained networks are non-random, these works are unable to capture the existence of adversarial examples in trained networks.

The result closest to ours was shown in Vardi, Yehudai, and Shamir [VYS22]. Similarly to our result, they used the KKT conditions of the maximum-margin problem in parameter space, in order to prove that gradient flow converges to non-robust two-layer ReLU networks under certain assumptions. More precisely, they considered a setting where the training dataset $\mathcal{S}$ consists of nearly-orthogonal points, and proved that every KKT point is non-robust w.r.t. $\mathcal{S}$. Namely, for every two-layer network that satisfies the KKT conditions of the maximum-margin problem, and every point $\mathbf{x}_i$ from $\mathcal{S}$, it is possible to flip the output's sign with a small perturbation. Their result has two main limitations: (1) It considers robustness w.r.t. the training data, while the more common setting in the literature considers robustness w.r.t. test data, as it is often more crucial to avoid adversarial perturbations in test examples; (2) Since they assume near orthogonality of the training data, the size of the dataset $\mathcal{S}$ must be smaller than the input dimension.[2] Thus, they considered a *high dimensional* setting. We note that high-dimensional settings often have a different generalization behavior than low-dimensional settings (e.g., overfitting can be *benign* in the high-dimensional setting, but harmful in a low-dimensional setting [KYS23]). Our result does not suffer from these limitations, since we consider robustness w.r.t. test data, and the size of our training dataset might be very large. In our results, we essentially require near orthogonality of the cluster means, as opposed to near orthogonality of the training dataset in their result.

Finally, in Bubeck, Li, and Nagaraj [BLN21] and Bubeck and Sellke [BS21], the authors proved (under certain assumptions) that overparameterization is necessary if one wants to interpolate training data using a neural network with a small Lipschitz constant. Namely, neural networks with a small number of parameters are not expressive enough to interpolate the training data while having a small

---

[2]They also give a version of their result, where instead of assuming this upper bound on the size of the dataset, they assume an upper bound on the number of points that attain the margin in the trained network, but it is not clear a priori when this assumption is likely to hold.

Lipschitz constant. These results suggest that overparameterization might be necessary for robustness. In this work, we show that even if the network is highly overparameterized, the implicit bias of the optimization method can prevent convergence to robust solutions.

## 2 Preliminaries

We use bold-face letters to denote vectors, e.g., $\mathbf{x} = (x_1, \ldots, x_d)$. For $\mathbf{x} \in \mathbb{R}^d$ we denote by $\|\mathbf{x}\|$ the Euclidean norm. We denote by $\mathbb{1}[\cdot]$ the indicator function, for example $\mathbb{1}[t \geq 5]$ equals 1 if $t \geq 5$ and 0 otherwise. We denote $\text{sign}(z) = 1$ if $z > 0$ and $-1$ otherwise. For an integer $d \geq 1$ we denote $[d] = \{1, \ldots, d\}$. For a set $A$ we denote by $\mathcal{U}(A)$ the uniform distribution over $A$. We denote by $\mathsf{N}(\mu, \sigma^2)$ the normal distribution with mean $\mu \in \mathbb{R}$ and variance $\sigma^2$, and by $\mathsf{N}(\boldsymbol{\mu}, \Sigma)$ the multivariate normal distribution with mean $\boldsymbol{\mu}$ and covariance matrix $\Sigma$. The identity matrix of size $d$ is denoted by $I_d$. We use standard asymptotic notation $\mathcal{O}(\cdot)$ and $\Omega(\cdot)$ to hide constant factors, and $\tilde{\mathcal{O}}(\cdot), \tilde{\Omega}(\cdot)$ to hide logarithmic factors. We use $\log$ for the logarithm with base 2 and $\ln$ for the natural logarithm.

In this work, we consider depth-2 ReLU neural networks. The ReLU activation function is defined by $\phi(z) = \max\{0, z\}$. Formally, a depth-2 network $\mathcal{N}_{\boldsymbol{\theta}}$ of width $m$ is parameterized by $\boldsymbol{\theta} = [\mathbf{w}_1, \ldots, \mathbf{w}_m, \mathbf{b}, \mathbf{v}]$ where $\mathbf{w}_i \in \mathbb{R}^d$ for all $i \in [m]$ and $\mathbf{b}, \mathbf{v} \in \mathbb{R}^m$, and for every input $\mathbf{x} \in \mathbb{R}^d$ we have
$$\mathcal{N}_{\boldsymbol{\theta}}(\mathbf{x}) = \sum_{j \in [m]} v_j \phi(\mathbf{w}_j^\top \mathbf{x} + b_j) .$$
We sometimes view $\boldsymbol{\theta}$ as the vector obtained by concatenating the vectors $\mathbf{w}_1, \ldots, \mathbf{w}_m, \mathbf{b}, \mathbf{v}$. Thus, $\|\boldsymbol{\theta}\|$ denotes the $\ell_2$ norm of the vector $\boldsymbol{\theta}$. We note that in this work we train both layers of the ReLU network.

We denote $\Phi(\boldsymbol{\theta}; \mathbf{x}) := \mathcal{N}_{\boldsymbol{\theta}}(\mathbf{x})$. We say that a network is *homogeneous* if there exists $L > 0$ such that for every $\alpha > 0$ and $\boldsymbol{\theta}, \mathbf{x}$ we have $\Phi(\alpha \boldsymbol{\theta}; \mathbf{x}) = \alpha^L \Phi(\boldsymbol{\theta}; \mathbf{x})$. Note that depth-2 ReLU networks as defined above are homogeneous (with $L = 2$).

We next define gradient flow and remind the reader of some recent results on the implicit bias of gradient flow in two-layer ReLU networks. Let $\mathcal{S} = \{(\mathbf{x}_i, y_i)\}_{i=1}^n \subseteq \mathbb{R}^d \times \{-1, 1\}$ be a binary classification training dataset. Let $\Phi(\boldsymbol{\theta}; \cdot) : \mathbb{R}^d \to \mathbb{R}$ be a neural network parameterized by $\boldsymbol{\theta}$. For a loss function $\ell : \mathbb{R} \to \mathbb{R}$ the *empirical loss* of $\Phi(\boldsymbol{\theta}; \cdot)$ on the dataset $\mathcal{S}$ is
$$\mathcal{L}(\boldsymbol{\theta}) := \tfrac{1}{n} \sum_{i=1}^n \ell(y_i \Phi(\boldsymbol{\theta}; \mathbf{x}_i)) . \tag{1}$$
We focus on the exponential loss $\ell(q) = e^{-q}$ and the logistic loss $\ell(q) = \log(1 + e^{-q})$.

We consider gradient flow on the objective given in Eq. (1). This setting captures the behavior of gradient descent with an infinitesimally small step size. Let $\boldsymbol{\theta}(t)$ be the trajectory of gradient flow. Starting from an initial point $\boldsymbol{\theta}(0)$, the dynamics of $\boldsymbol{\theta}(t)$ is given by the differential equation $\frac{d\boldsymbol{\theta}(t)}{dt} \in -\partial^\circ \mathcal{L}(\boldsymbol{\theta}(t))$. Here, $\partial^\circ$ denotes the *Clarke subdifferential* [Cla+08], which is a generalization of the derivative for non-differentiable functions.

We now remind the reader of a recent result concerning the implicit bias of gradient flow over the exponential and logistic losses for homogeneous neural networks. Note that since homogeneous networks satisfy $\text{sign}(\Phi(\alpha\boldsymbol{\theta}; \mathbf{x})) = \text{sign}(\Phi(\boldsymbol{\theta}; \mathbf{x}))$ for any $\alpha > 0$, the sign of the network output of homogeneous networks depends only on the direction of the parameters $\boldsymbol{\theta}$. The following theorem provides a characterization of the implicit bias of gradient flow by showing that the trajectory of the weights $\boldsymbol{\theta}(t)$ *converge in direction* to a first-order stationary point of a particular constrained optimization problem, where $\boldsymbol{\theta}$ *converges in direction* to $\tilde{\boldsymbol{\theta}}$ means $\lim_{t\to\infty} \frac{\boldsymbol{\theta}(t)}{\|\boldsymbol{\theta}(t)\|} = \frac{\tilde{\boldsymbol{\theta}}}{\|\tilde{\boldsymbol{\theta}}\|}$. Note that since ReLU networks are non-smooth, the first-order stationarity conditions (i.e., the Karush–Kuhn–Tucker conditions, or KKT conditions for short) are defined using the Clarke subdifferential (see Lyu and Li [LL20] and Dutta et al. [Dut+13] for more details on the KKT conditions in non-smooth optimization problems).

**Theorem 2.1** (Paraphrased from Lyu and Li [LL20] and Ji and Telgarsky [JT20])**.** *Let $\Phi(\boldsymbol{\theta}; \cdot)$ be a homogeneous ReLU neural network parameterized by $\boldsymbol{\theta}$. Consider minimizing either the exponential or the logistic loss over a binary classification dataset $\{(\mathbf{x}_i, y_i)\}_{i=1}^n$ using gradient flow. Assume that there exists time $t_0$ such that $\mathcal{L}(\boldsymbol{\theta}(t_0)) < \frac{1}{n}$ (and thus $y_i \Phi(\boldsymbol{\theta}(t_0); \mathbf{x}_i) > 0$ for every $\mathbf{x}_i$). Then, gradient flow converges in direction to a first-order stationary point (KKT point) of the following*

*maximum margin problem in parameter space:*

$$\min_{\boldsymbol{\theta}} \tfrac{1}{2} \|\boldsymbol{\theta}\|^2 \quad s.t. \quad \forall i \in [n] \ \ y_i \Phi(\boldsymbol{\theta}; \mathbf{x}_i) \geq 1 \ . \tag{2}$$

*Moreover, $\mathcal{L}(\boldsymbol{\theta}(t)) \to 0$ and $\|\boldsymbol{\theta}(t)\| \to \infty$ as $t \to \infty$.*

Theorem 2.1 gives a characterization of the implicit bias of gradient flow with the exponential and the logistic loss for homogeneous ReLU networks. Note that the theorem makes no assumption on the initialization, training data, or number of parameters in the network; the only requirement is that the network is homogeneous and that at some time point in the gradient flow trajectory, the network is able to achieve small training loss. The theorem shows that although there are many ways to configure the network parameters to achieve small training loss (via overparameterization), gradient flow only converges (in direction) to networks which satisfy the KKT conditions of Problem (2). It is important to note that satisfaction of the KKT conditions is not sufficient for global optimality of the constrained optimization problem [VSS22]. We further note that if the training data are sampled i.i.d. from a distribution with label noise (e.g., a class-conditional Gaussian mixture model, or a distribution where labels $y_i$ are flipped to $-y_i$ with some nonzero probability), networks which have parameters that are feasible w.r.t. the constraints of Problem (2) have overfit to noise, and understanding the generalization behavior of even globally optimal solutions to Problem (2) in this setting is the subject of significant research [Mon+19; CL21; Fre+23a].

Finally, we introduce the distributional setting that we consider. We consider a distribution $\mathcal{D}_{\text{clusters}}$ on $\mathbb{R}^d \times \{-1, 1\}$ that consists of $k$ clusters with means $\boldsymbol{\mu}^{(1)}, \dots, \boldsymbol{\mu}^{(k)} \in \mathbb{R}^d$ and covariance $\sigma^2 I_d$ (i.e., a Gaussian mixture model), such that the examples in the $j$-th cluster are labeled by $y^{(j)} \in \{-1, 1\}$. More formally, $(\mathbf{x}, y) \sim \mathcal{D}_{\text{clusters}}$ is generated as follows: we draw $j \sim \mathcal{U}([k])$ and $\mathbf{x} \sim \mathsf{N}(\boldsymbol{\mu}^{(j)}, \sigma^2 I_d)$, and set $y = y^{(j)}$. We assume that there exist $i, j \in [k]$ with $y^{(i)} \neq y^{(j)}$. Moreover, we assume the following:

**Assumption 2.2.** *We have:*

- $\left\|\boldsymbol{\mu}^{(j)}\right\| = \sqrt{d}$ *for all $j \in [k]$.*

- $0 < \sigma \leq 1$.

- $k \left( \max_{i \neq j} |\langle \boldsymbol{\mu}^{(i)}, \boldsymbol{\mu}^{(j)} \rangle| + 4\sigma\sqrt{d}\ln(d) + 1 \right) \leq \frac{d - 4\sigma\sqrt{d}\ln(d) + 1}{10}$.

**Example 1.** *Below we provide simple examples of settings that satisfy the assumption:*

- *Suppose that the cluster means satisfy $|\langle \boldsymbol{\mu}^{(i)}, \boldsymbol{\mu}^{(j)} \rangle| = \tilde{\mathcal{O}}(\sqrt{d})$ for every $i \neq j$. This condition holds, e.g., if we choose each cluster mean i.i.d. from the uniform distribution on the sphere $\sqrt{d} \cdot \mathbb{S}^{d-1}$ (see, e.g., Vardi, Yehudai, and Shamir [VYS22, Lemma 3.1]). Let $\sigma = 1$, namely, each cluster has a radius of roughly $\sqrt{d}$. Then, the assumption can be satisfied by choosing $k = \tilde{\mathcal{O}}(\sqrt{d})$.*

- *Suppose that the cluster means are exactly orthogonal (i.e., $\langle \boldsymbol{\mu}^{(i)}, \boldsymbol{\mu}^{(j)} \rangle = 0$ for all $i \neq j$), and $\sigma = 1/\sqrt{d}$. Then, the assumption can be satisfied by choosing $k = \tilde{\mathcal{O}}(d)$.*

- *If the number of clusters is $k = \tilde{\mathcal{O}}(1)$, then the assumption may hold even where $\max_{i \neq j} |\langle \boldsymbol{\mu}^{(i)}, \boldsymbol{\mu}^{(j)} \rangle| = \tilde{\Theta}(d)$ (for any $0 < \sigma \leq 1$).*

A few remarks are in order. First, the assumption that $\left\|\boldsymbol{\mu}^{(j)}\right\|$ is exactly $\sqrt{d}$ is for convenience, and we note that it may be relaxed (to have all cluster means approximately of the same norm) without affecting our results significantly. Note that in the case where $\sigma = 1$, the radius of each cluster is roughly of the same magnitude as the cluster mean. Second, we assume for convenience that the noise (i.e., the deviation from the cluster's mean) is drawn from a Gaussian distribution with covariance matrix $\sigma^2 I_d$. However, we note that this assumption can be generalized to any distribution $\mathcal{D}_{\text{noise}}$ such that for every unit vector $\mathbf{e}$ the noise $\boldsymbol{\xi} \sim \mathcal{D}_{\text{noise}}$ satisfies w.h.p. that $\langle \boldsymbol{\xi}, \mathbf{e} \rangle = \tilde{\mathcal{O}}(1)$ and $\|\boldsymbol{\xi}\| = \tilde{\mathcal{O}}(\sqrt{d})$. This property holds, e.g., for a $d$-dimensional Gaussian distribution $\mathsf{N}(0, \Sigma)$, where $\text{tr}[\Sigma] = d$ and $\|\Sigma\|_2 = O(1)$ (see Frei et al. [Fre+23b, Lemma 3.3]), and more generally for a class of sub-Gaussian distributions (see Hu et al. [Hu+20, Claim 3.1]). Third, note that the third part of Assumption 2.2 essentially requires that the number of clusters $k$ cannot be too large

and the correlations between cluster means cannot be too large. Finally, we remark that when $k$ is small, our results may be extended to the case where $\sigma > 1$. For example, if $k = \tilde{\mathcal{O}}(1)$ and $\max_{i \neq j} |\langle \boldsymbol{\mu}^{(i)}, \boldsymbol{\mu}^{(j)} \rangle| = \tilde{\mathcal{O}}(\sqrt{d})$, our generalization result (Theorem 3.1) can be extended to the case where $\sigma = \tilde{\mathcal{O}}(d^{1/8})$. We preferred to avoid handling $\sigma > 1$ in order to simplify the proofs.

Moreover, it is worth noting that Assumption 2.2 implies that the data is w.h.p. linearly separable (see Lemma 2.1 below, and a proof in Appendix B). However, in this work we consider learning using overparameterized ReLU networks, and it is not obvious a priori that gradient methods do not harmfully overfit in this case. Indeed, it has been shown that ReLU networks trained by gradient descent can interpolate training data and fail to generalize well in some distributional settings [Kou+23].

**Lemma 2.1.** *Let* $\mathbf{u} = \sum_{q \in [k]} y^{(q)} \boldsymbol{\mu}^{(q)}$. *Then, with probability at least* $1 - 2d^{1-\ln(d)/2} = 1 - o_d(1)$ *over* $(\mathbf{x}, y) \sim \mathcal{D}_{clusters}$, *we have* $y = \mathrm{sign}(\mathbf{u}^\top \mathbf{x})$.

## 3 Generalization

In this section, we show that under our assumptions on the distribution $\mathcal{D}_{\text{clusters}}$, gradient flow does not harmfully overfit. Namely, even if the learned network is highly overparameterized, the implicit bias of gradient flow guarantees convergence to a solution that generalizes well. Moreover, we show that the sample complexity is optimal. The main result of this section is stated in the following theorem:

**Theorem 3.1.** *Let* $\epsilon, \delta \in (0, 1)$. *Let* $\mathcal{S} = \{(\mathbf{x}_i, y_i)\}_{i=1}^n \subseteq \mathbb{R}^d \times \{-1, 1\}$ *be a training set drawn i.i.d. from the distribution* $\mathcal{D}_{clusters}$, *where* $n \geq k \ln^2(d)$. *Let* $\mathcal{N}_{\boldsymbol{\theta}}$ *be a depth-2 ReLU network such that* $\boldsymbol{\theta} = [\mathbf{w}_1, \ldots, \mathbf{w}_m, \mathbf{b}, \mathbf{v}]$ *is a KKT point of Problem (2). Provided* $d$ *is sufficiently large such that* $\delta^{-1} \leq \frac{1}{3} d^{\ln(d)-1}$ *and* $n \leq \min\left\{ \sqrt{\frac{\delta}{3}} \cdot e^{d/32}, \frac{\sqrt{\delta}}{3} \cdot d^{\ln(d)/4}, \frac{\epsilon}{4} \cdot d^{\ln(d)/2} \right\}$, *then with probability at least* $1 - \delta$ *over* $\mathcal{S}$, *we have*

$$\Pr_{(\mathbf{x}, y) \sim \mathcal{D}_{clusters}} \left[ y \mathcal{N}_{\boldsymbol{\theta}}(\mathbf{x}) \leq 0 \right] \leq \epsilon \,.$$

The sample complexity requirement in Theorem 3.1 is $n = \tilde{\Omega}(k)$. Essentially, it requires that the dataset $\mathcal{S}$ will include at least one example from each cluster. Clearly, any learning algorithm cannot perform well on unseen clusters. Hence the sample complexity requirement in the theorem is tight (up to log factors).

The assumptions in Theorem 3.1 include upper bounds on $\delta^{-1}$ and $n$. Note that the expressions in these upper bounds are super-polynomial in $d$, and in particular if $n, \delta^{-1}, \epsilon^{-1} = \mathrm{poly}(d)$, then these assumptions hold for a sufficiently large $d$. Admittedly, enforcing an upper bound on the training dataset's size is uncommon in generalization results. However, if $n$ is exponential in $d$, it is not hard to see that there will be clusters which have both positive and negative examples within radius $\sigma$ of the cluster center, essentially introducing a form of label noise to the problem. Since KKT points of Problem (2) interpolate the training data, this would imply that the network has interpolated training data with label noise—in other words, it has 'overfit' to noise. Understanding the generalization behavior of interpolating neural networks in the presence of label noise is a very technically challenging problem for which much is unknown, especially if one seeks to understand this by only relying upon the properties of KKT conditions for margin maximization. It is noteworthy that all existing non-vacuous generalization bounds for interpolating nonlinear neural networks in the presence of label noise require $n < d$ [FCB22; Cao+22; XG23; Fre+23a; Kou+23].

Combining Theorem 3.1 with Theorem 2.1, we conclude that w.h.p. over a training dataset of size $n \geq k \ln^2(d)$ (and under some additional mild requirements), if gradient flow reaches empirical loss smaller than $\frac{1}{n}$, then it converges in direction to a neural network that generalizes well. This result is width-independent, thus, it holds irrespective of the network width. Specifically, even if the network is highly overparameterized, the implicit bias of gradient flow prevents harmful overfitting. Moreover, the result does not depend directly on the initialization of gradient flow. That is, it holds whenever gradient flow reaches small empirical loss after some finite time. Thus, by relying on the KKT conditions of the max-margin problem instead of analyzing the full gradient flow trajectory, we can prove generalization without the need to prove convergence.

## 3.1 Proof idea

The proof of Theorem 3.1 is given in Appendix A. Here we discuss the high-level approach. Let $\boldsymbol{\theta} = [\mathbf{w}_1, \ldots, \mathbf{w}_m, \mathbf{b}, \mathbf{v}]$ be a KKT point of Problem (2). Thus, we have $\mathcal{N}_{\boldsymbol{\theta}}(\mathbf{x}) = \sum_{j \in [m]} v_j \phi(\mathbf{w}_j^\top \mathbf{x} + b_j)$. Since $\boldsymbol{\theta}$ satisfies the KKT conditions of Problem (2), then there are $\lambda_1, \ldots, \lambda_n$ such that for every $j \in [m]$ we have

$$\mathbf{w}_j = \sum_{i \in [n]} \lambda_i \nabla_{\mathbf{w}_j} \left( y_i \mathcal{N}_{\boldsymbol{\theta}}(\mathbf{x}_i) \right) = \sum_{i \in [n]} \lambda_i y_i v_j \phi'_{i,j} \mathbf{x}_i , \tag{3}$$

where $\phi'_{i,j}$ is a subgradient of $\phi$ at $\mathbf{w}_j^\top \mathbf{x}_i + b_j$, i.e., if $\mathbf{w}_j^\top \mathbf{x}_i + b_j \neq 0$ then $\phi'_{i,j} = \mathbb{1}[\mathbf{w}_j^\top \mathbf{x}_i + b_j \geq 0]$, and otherwise $\phi'_{i,j}$ is some value in $[0, 1]$. Also we have $\lambda_i \geq 0$ for all $i$, and $\lambda_i = 0$ if $y_i \mathcal{N}_{\boldsymbol{\theta}}(\mathbf{x}_i) \neq 1$. Likewise, we have

$$b_j = \sum_{i \in [n]} \lambda_i \nabla_{b_j} \left( y_i \mathcal{N}_{\boldsymbol{\theta}}(\mathbf{x}_i) \right) = \sum_{i \in [n]} \lambda_i y_i v_j \phi'_{i,j} . \tag{4}$$

In the proof, using a careful analysis of Eq. (3) and (4) we show that w.h.p. $\mathcal{N}_{\boldsymbol{\theta}}$ classifies correctly a fresh example. More precisely, the main argument can be described as follows. We denote $J := [m]$, $J_+ := \{j \in J : v_j > 0\}$, and $J_- := \{j \in J : v_j < 0\}$. Moreover, we denote $I := [n]$ and $Q := [k]$. For $q \in Q$ we denote $I^{(q)} = \{i \in I : \mathbf{x}_i \text{ is in cluster } q\}$. Consider the network's output for an input $\mathbf{x}$ from cluster $r \in Q$ with $y^{(r)} = 1$. Since $\mathcal{N}_{\boldsymbol{\theta}}(\mathbf{x}) = \sum_{j \in J_+} v_j \phi(\mathbf{w}_j^\top \mathbf{x} + b_j) + \sum_{j \in J_-} v_j \phi(\mathbf{w}_j^\top \mathbf{x} + b_j)$ and $\phi(z) \geq z$, we have

$$\mathcal{N}_{\boldsymbol{\theta}}(\mathbf{x}) \geq \sum_{j \in J_+} v_j (\mathbf{w}_j^\top \mathbf{x} + b_j) + \sum_{j \in J_-} v_j \phi(\mathbf{w}_j^\top \mathbf{x} + b_j). \tag{5}$$

This suggests the following possibility: if we can ensure that $\sum_{j \in J_+} v_j(\mathbf{w}_j^\top \mathbf{x} + b_j)$ is large and positive while $\sum_{j \in J_-} v_j \phi(\mathbf{w}_j^\top \mathbf{x} + b_j)$ is not too negative, then the network will accurately classify the example $\mathbf{x}$. Using Eq. (3) and (4) and that $y^{(r)} = 1$ (so $y_i = 1$ for $i \in I^{(r)}$), the first term in the above decomposition is equal to

$$\sum_{j \in J_+} v_j(\mathbf{w}_j^\top \mathbf{x} + b_j) = \sum_{j \in J_+} v_j \left[ \sum_{i \in I} \lambda_i y_i v_j \phi'_{i,j}(\mathbf{x}_i^\top \mathbf{x} + 1) \right]$$

$$= \sum_{j \in J_+} \left[ \left( \sum_{i \in I^{(r)}} \lambda_i v_j^2 \phi'_{i,j}(\mathbf{x}_i^\top \mathbf{x} + 1) \right) + \sum_{q \in Q \setminus \{r\}} \sum_{i \in I^{(q)}} \lambda_i y_i v_j^2 \phi'_{i,j}(\mathbf{x}_i^\top \mathbf{x} + 1) \right]$$

$$\geq \left( \sum_{i \in I^{(r)}} \sum_{j \in J_+} \lambda_i v_j^2 \phi'_{i,j}(\mathbf{x}_i^\top \mathbf{x} + 1) \right) - \sum_{q \in Q \setminus \{r\}} \sum_{i \in I^{(q)}} \sum_{j \in J_+} \lambda_i v_j^2 \phi'_{i,j} |\mathbf{x}_i^\top \mathbf{x} + 1| .$$

Since $\mathbf{x}$ comes from cluster $r$ and the clusters are nearly orthogonal, the pairwise correlations $\mathbf{x}_i^\top \mathbf{x}$ will be large and positive when $i \in I^{(r)}$ but will be small in magnitude when $i \in I^{(q)}$ for $q \neq r$. Thus, we can hope that this term will be large and positive if we can show that the quantity $\sum_{i \in I^{(r)}} \sum_{j \in J^+} \lambda_i v_j^2 \phi'_{i,j}$ is not too small relative to the quantity $\sum_{q \in Q \setminus \{r\}} \sum_{i \in I^{(q)}} \sum_{j \in J_+} \lambda_i v_j^2 \phi'_{i,j}$. By similar arguments, in order to show the second term in Eq. (5) is not too negative, we need to understand how the quantity $\sum_{i \in I^{(q)}} \sum_{j \in J_-} \lambda_i v_j^2 \phi'_{i,j}$ varies across different clusters $q \in Q$. Hence, in the proof we analyze how the quantities $\sum_{i \in I^{(q)}} \sum_{j \in J_+} \lambda_i v_j^2 \phi'_{i,j}$, $\sum_{i \in I^{(q)}} \sum_{j \in J_-} \lambda_i v_j^2 \phi'_{i,j}$ relate to each other for different clusters $q \in Q$, and show that these quantities are all of the same order. Then, we conclude that w.h.p. $\mathbf{x}$ is classified correctly.

## 4 Robustness

We begin by introducing the definition of $R(\cdot)$-robustness.

**Definition 4.1.** *Given some function $R(\cdot)$, we say that a neural network $\mathcal{N}_{\boldsymbol{\theta}}$ is $R(d)$-robust w.r.t. a distribution $\mathcal{D}_{\mathbf{x}}$ over $\mathbb{R}^d$ if for every $r = o(R(d))$, with probability $1 - o_d(1)$ over $\mathbf{x} \sim \mathcal{D}_{\mathbf{x}}$, for every $\mathbf{x}' \in \mathbb{R}^d$ with $\|\mathbf{x} - \mathbf{x}'\| \leq r$ we have $\text{sign}(\mathcal{N}_{\boldsymbol{\theta}}(\mathbf{x}')) = \text{sign}(\mathcal{N}_{\boldsymbol{\theta}}(\mathbf{x}))$.*

Thus, a neural net $\mathcal{N}_{\boldsymbol{\theta}}$ is $R(d)$-robust if changing the label of an example cannot be done with a perturbation of size $o(R(d))$. Note that we consider here $\ell_2$ perturbations.

For the distribution $\mathcal{D}_{\text{clusters}}$ under consideration, it is straightforward to show that classifiers cannot be $R(d)$-robust if $R(d) = \omega(\sqrt{d})$: since the distance between examples in different clusters is w.h.p.

$\mathcal{O}(\sqrt{d})$, it is clearly possible to flip the sign of an example with a perturbation of size $\mathcal{O}(\sqrt{d})$. In particular, the best we can hope for is $\sqrt{d}$-robustness. In the following theorem, we show that there exist two-layer ReLU networks which can both achieve small test error and the optimal level of $\sqrt{d}$-robustness.

**Theorem 4.1.** *For every $r \geq k$, there exists a depth-2 ReLU network $\mathcal{N} : \mathbb{R}^d \to \mathbb{R}$ of width $r$ such that for $(\mathbf{x}, y) \sim \mathcal{D}_{clusters}$, with probability at least $1 - d^{-\omega_d(1)}$ we have $y\mathcal{N}(\mathbf{x}) \geq 1$, and flipping the sign of the output requires a perturbation of size larger than $\frac{\sqrt{d}}{8}$ (for a sufficiently large $d$). Thus, $\mathcal{N}$ classifies the data correctly w.h.p., and it is $\sqrt{d}$-robust w.r.t. $\mathcal{D}_{\mathbf{x}}$.*

Thus, we see that $\sqrt{d}$-robust networks exist. In the following theorem, we show that the implicit bias of gradient flow constrains the level of robustness of *trained* networks whenever the number of clusters $k$ is large.

**Theorem 4.2.** *Let $\epsilon, \delta \in (0, 1)$. Let $\mathcal{S} = \{(\mathbf{x}_i, y_i)\}_{i=1}^n \subseteq \mathbb{R}^d \times \{-1, 1\}$ be a training set drawn i.i.d. from the distribution $\mathcal{D}_{clusters}$, where $n \geq k \ln^2(d)$. We denote $Q_+ = \{q \in [k] : y^{(q)} = 1\}$ and $Q_- = \{q \in [k] : y^{(q)} = -1\}$, and assume that $\min\left\{\frac{|Q_+|}{k}, \frac{|Q_-|}{k}\right\} \geq c$ for some $c > 0$. Let $\mathcal{N}_{\boldsymbol{\theta}}$ be a depth-2 ReLU network such that $\boldsymbol{\theta} = [\mathbf{w}_1, \ldots, \mathbf{w}_m, \mathbf{b}, \mathbf{v}]$ is a KKT point of Problem (2). Provided $d$ is sufficiently large such that $\delta^{-1} \leq \frac{1}{3} d^{\ln(d)-1}$ and $n \leq \min\left\{\sqrt{\frac{\delta}{3}} \cdot e^{d/32}, \frac{\sqrt{\delta}}{3} \cdot d^{\ln(d)/4}, \frac{\epsilon}{4} \cdot d^{\ln(d)/2}\right\}$, with probability at least $1 - \delta$ over $\mathcal{S}$, there is a vector $\mathbf{z} = \eta \cdot \sum_{j \in [k]} y^{(j)} \boldsymbol{\mu}^{(j)}$ with $\eta > 0$ and $\|\mathbf{z}\| \leq \mathcal{O}\left(\sqrt{d/c^2 k}\right)$, such that*

$$\Pr_{(\mathbf{x}, y) \sim \mathcal{D}_{clusters}} [\text{sign}(\mathcal{N}_{\boldsymbol{\theta}}(\mathbf{x})) \neq \text{sign}(\mathcal{N}_{\boldsymbol{\theta}}(\mathbf{x} - y\mathbf{z}))] \geq 1 - \epsilon .$$

Note that the expressions in the upper bounds on $n$ and $\delta^{-1}$ are super-polynomial in $d$, and hence these requirements are mild (e.g., they hold for a sufficiently large $d$ when $n, \delta^{-1}, \epsilon^{-1} = \text{poly}(d)$). As we mentioned in the discussion following Theorem 3.1, we believe removing the requirement for an upper bound on $n$ would be highly nontrivial.

Theorem 4.2 implies that if $c^2 k = \omega_d(1)$, then w.h.p. over the training dataset, every KKT point of Problem (2) is not $\sqrt{d}$-robust. Specifically, if $c$ is constant, namely, at least a constant fraction of the clusters have positive labels and a constant fraction of the clusters have negative labels, then the network is not $\sqrt{d}$-robust if $k = \omega_d(1)$. Recall that by Theorem 3.1, we also have w.h.p. that every KKT point generalizes well. Overall, combining Theorems 2.1, 3.1, 4.1, and 4.2, we conclude that for $c^2 k = \omega_d(1)$, w.h.p. over a training dataset of size $n \geq k \ln^2(d)$, if gradient flow reaches empirical loss smaller than $\frac{1}{n}$, then it converges in direction to a neural network that generalizes well but is not $\sqrt{d}$-robust, even though there exist $\sqrt{d}$-robust networks that generalize well. Thus, in our setting, there is bias towards solutions that generalize well but are non-robust.

**Example 2.** *Consider the setting from the first item of Example 1. Thus, the cluster means satisfy $|\langle \boldsymbol{\mu}^{(i)}, \boldsymbol{\mu}^{(j)} \rangle| = \tilde{\mathcal{O}}(\sqrt{d})$ for every $i \neq j$, and we have $\sigma = 1$ and $k = \tilde{\Theta}(\sqrt{d})$. Suppose that $c = \Theta(1)$, namely, there is at least a constant fraction of clusters with each label $\{-1, 1\}$. Then, the adversarial perturbation $\mathbf{z}$ from Theorem 4.2 satisfies $\|\mathbf{z}\| = \mathcal{O}\left(\sqrt{d/k}\right) = \tilde{\mathcal{O}}\left(d^{1/4}\right) = o(\sqrt{d})$.*

Similarly to our discussion after Theorem 3.1, we note that Theorem 4.2 is width-independent, i.e., it holds irrespective of the network width. It implies that we cannot hope to obtain a robust solution by choosing an appropriate width for the trained network. As we discussed in the related work section, Bubeck, Li, and Nagaraj [BLN21] and Bubeck and Sellke [BS21] considered the expressive power of neural networks, and showed that overparameterization might be necessary for robustness. By Theorem 4.2, even when the network is overparameterized, the implicit bias of the optimization method can prevent convergence to robust solutions. Moreover, our result does not depend directly on the initialization of gradient flow. Recall that by Theorem 2.1 if gradient flow reaches small empirical loss then it converges in direction to a KKT point of Problem (2). Hence our result holds whenever gradient flow reaches a small empirical loss.

Note that in Theorem 4.2, the adversarial perturbation does not depend on the input (up to sign). It corresponds to the well-known empirical phenomenon of *universal adversarial perturbations*, where one can find a single perturbation that simultaneously flips the label of many inputs (cf. [Moo+17;

Zha+21]). Moreover, the same perturbation applies to all depth-2 networks to which gradient flow might converge (i.e., all KKT points). It corresponds to the well-known empirical phenomenon of *transferability* in adversarial examples, where one can find perturbations that simultaneously flip the labels of many different trained networks (cf. [Liu+17; AM18]).

It is worth noting that Theorems 3.1 and 4.2 demonstrate that trained neural networks exhibit different properties than the 1-nearest-neighbour learning rule, irrespective of the number of parameters in the network. For example, consider the case where $\sigma = \frac{1}{\sqrt{d}}$, namely, the examples of each cluster are concentrated within a ball of radius $O(1)$ around its mean. Then, the distance between every pair of points from the same cluster is $O(1)$, and the distance between points from different clusters is $\Omega(\sqrt{d})$. In this setting, both the 1-nearest-neighbour classifier and the trained neural network will classify a fresh example correctly w.h.p., but in the 1-nearest-neighbour classifier flipping the output's sign will require a perturbation of size $\Omega(\sqrt{d})$, while in the neural network a much smaller perturbation will suffice.

Finally, we remark that in the limit $\sigma \to 0$, we get a distribution supported on $\boldsymbol{\mu}^{(1)}, \dots, \boldsymbol{\mu}^{(k)}$. Then, a training dataset of size $n \geq k \ln^2(d)$ will contain w.h.p. all examples in the support, and hence robustness w.r.t. test data is equivalent to robustness w.r.t. the training data. In this case, we recover the results of Vardi, Yehudai, and Shamir [VYS22] which characterized the non-robustness of KKT points of ReLU networks trained on nearly orthogonal training data. In particular, our Theorem 4.2 is a strict generalization of their Theorem 4.1.

## 4.1  Proof ideas

Here we discuss the main ideas in the proofs of Theorem 4.1 and 4.2 (see Appendices C and D for the formal proofs).

The proof of Theorem 4.1 follows by the following simple construction. The robust network includes $k$ neurons, each corresponding to a single cluster. That is, we have $\mathcal{N}(\mathbf{x}) = \sum_{j=1}^{k} v_j \sigma(\mathbf{w}^\top \mathbf{x} + b_j)$, where $v_j = y^{(j)}$, $\mathbf{w}_j = \frac{4\boldsymbol{\mu}^{(j)}}{d}$, and $b_j = -2$. Note that the $j$-th neuron points at the direction of the $j$-th cluster and has a negative bias term, such that the neuron is active on points from the $j$-th cluster, and inactive on points from the other clusters. Then, given a fresh example $(\mathbf{x}, y) \sim \mathcal{D}_{\text{clusters}}$, we show that the network classifies it correctly w.h.p. with margin at least 1. Also, there is w.h.p exactly one neuron that is active on $\mathbf{x}$, and hence the gradient of the network w.r.t. the input is affected only by this neuron and is of size $\mathcal{O}(1/\sqrt{d})$. Therefore, we need a perturbation of size $\Omega(\sqrt{d})$ in order to flip the output's sign.

The intuition for Theorem 4.2 can be described as follows. Recall that in our construction of a robust network above, an example $(\mathbf{x}, y) \sim \mathcal{D}_{\text{clusters}}$ is w.h.p. in an active region of exactly one neuron, and hence in the neighborhood of $\mathbf{x}$ the output of the network is sensitive only to perturbations in the direction of that neuron. Now, consider the linear model $\mathbf{x} \mapsto \mathbf{w}^\top \mathbf{x}$, where $\mathbf{w} = \sum_{q=1}^{k} \frac{1}{d} y^{(q)} \boldsymbol{\mu}^{(q)}$. It is not hard to verify that for $(\mathbf{x}, y) \sim \mathcal{D}_{\text{clusters}}$ we have w.h.p. that $0 < y\mathbf{w}^\top \mathbf{x} \leq \mathcal{O}(1)$. Moreover, the gradient of this linear predictor is of size $\|\mathbf{w}\| = \Omega(\sqrt{k/d})$. Hence, we can flip the output's sign with a perturbation of size $\mathcal{O}(\sqrt{d/k})$. Thus, the linear classifier is non-robust if $k = \omega_d(1)$. Intuitively, the difference between our robust ReLU network and the non-robust linear classifier is the fact that in the neighborhood of $\mathbf{x}$ the robust network is sensitive only to perturbations in the direction of one cluster, while the linear classifier is sensitive to perturbations in the directions of all $k$ clusters. In the proof, we analyze ReLU networks which are KKT points of Problem (2), and show that although these ReLU networks are non-linear, they are still sensitive to perturbations in the directions of all $k$ clusters, similarly to the above linear classifier. The formal proof follows by a careful analysis of the KKT conditions of Problem (2), given in Eq. (3) and (4).

We remark that in the proof of Theorem 4.2 we use some technical ideas from Vardi, Yehudai, and Shamir [VYS22]. However, there are significant differences between the two settings. For example, they assume that the training data are nearly orthogonal, which only holds when the dimension is large relative to the number of samples; thus, it is unclear whether the existence of small adversarial perturbations in their setting is due to the high-dimensionality of the data or if a similar phenomenon exists in the more common $n > d$ setting. At a more technical level, their proof relies on showing that

in a KKT point all inputs must lie exactly on the margin, while in our setting they are not guaranteed to lie exactly on the margin.

## 5  Discussion

In this paper, we considered clustered data, and showed that gradient flow in two-layer ReLU networks does not harmfully overfit, but also hinders robustness. Our results follow by analyzing the KKT points of the max-margin problem in parameter space. In our distributional setting, the clusters are well-separated, and hence there exist robust classifiers, which allows us to consider the effect of the implicit bias of gradient flow on both generalization and robustness. Understanding generalization and robustness in additional data distributions and neural network architectures is a challenging but important question. As a possible next step, it would be interesting to study whether the approach used in this paper can be extended to the following data distributions:

First, our assumption on the data distribution (Assumption 2.2) implies that the number of clusters cannot be too large, and as a result the data is linearly separable (Lemma 2.1). We conjecture that our results hold even for a significantly larger number of clusters, such that the data is not linearly separable.

Second, it would be interesting to understand whether our generalization result holds for linearly separable data distributions that are not clustered. That is, given a distribution that is linearly separable with some margin $\gamma > 0$ and a training dataset that is large enough to allow learning with a max-margin linear classifier, are there KKT points of the max-margin problem for two-layer ReLU networks that do not generalize well? In other words, do ReLU networks that satisfy the KKT conditions generalize at least as well as max-margin linear classifiers?

## Acknowledgments and Disclosure of Funding

SF, GV, PB, and NS acknowledge the support of the NSF and the Simons Foundation for the Collaboration on the Theoretical Foundations of Deep Learning through awards DMS-2031883 and #814639, and of the NSF through grant DMS-2023505.

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
