# A   Proof of Theorem 3.1

We will prove the following theorem:

**Theorem A.1.** *Let $\mathcal{S} = \{(\mathbf{x}_i, y_i)\}_{i=1}^n \subseteq \mathbb{R}^d \times \{-1, 1\}$ be a training set drawn i.i.d. from the distribution $\mathcal{D}_{clusters}$, where $n \geq k \ln^2(d)$. Let $\mathcal{N}_{\boldsymbol{\theta}}$ be a depth-2 ReLU network such that $\boldsymbol{\theta} = [\mathbf{w}_1, \ldots, \mathbf{w}_m, \mathbf{b}, \mathbf{v}]$ is a KKT point of Problem (2). Then, with probability at least*

$$1 - \left( 3n^2 d^{-\frac{\ln(d)}{2}} + n^2 e^{-d/16} + d^{1-\ln(d)} \right)$$

*over $\mathcal{S}$, we have*

$$\Pr_{(\mathbf{x}, y) \sim \mathcal{D}_{clusters}} [y \mathcal{N}_{\boldsymbol{\theta}}(\mathbf{x})) \leq 0] \leq 4n d^{-\frac{\ln(d)}{2}} \ .$$

It is easy to verify that Theorem A.1 implies Theorem 3.1. Indeed, if $\frac{1}{\delta} \leq \frac{1}{3} d^{\ln(d)-1}$ and

$$n \leq \min \left\{ \sqrt{\frac{\delta}{3}} \cdot e^{d/32}, \frac{\sqrt{\delta}}{3} \cdot d^{\ln(d)/4}, \frac{\epsilon}{4} \cdot d^{\ln(d)/2} \right\} \ ,$$

then we have:

1.
$$3n^2 d^{-\frac{\ln(d)}{2}} \leq 3 \left( \frac{\sqrt{\delta}}{3} d^{\ln(d)/4} \right)^2 d^{-\ln(d)/2} = 3 \cdot \frac{\delta}{9} \cdot d^{\ln(d)/2} d^{-\ln(d)/2} = \frac{\delta}{3} \ .$$

2.
$$n^2 e^{-d/16} \leq \left( \sqrt{\frac{\delta}{3}} \cdot e^{d/32} \right)^2 e^{-d/16} = \frac{\delta}{3} \ .$$

3.
$$d^{1-\ln(d)} \leq \frac{\delta}{3} \ .$$

4.
$$4n d^{-\frac{\ln(d)}{2}} \leq 4 \cdot \frac{\epsilon}{4} \cdot d^{\ln(d)/2} \cdot d^{-\ln(d)/2} = \epsilon \ .$$

Hence, under the above assumptions on $\delta$ and $n$, Theorem A.1 implies that with probability at least $1 - \delta$ over $\mathcal{S}$, we have $\Pr_{(\mathbf{x}, y) \sim \mathcal{D}_{clusters}} [y \mathcal{N}_{\boldsymbol{\theta}}(\mathbf{x})) \leq 0] \leq \epsilon$.

We now turn to prove Theorem A.1. The high-level idea for the proof is as follows. First, we show that the training dataset is sufficiently "nice" with high probability, in the sense that samples within each cluster are highly correlated while samples in orthogonal clusters are nearly orthogonal (see properties (P1) through (P6) below). This analysis appears in Section A.1. We then show that datasets with "nice" properties impose a number of structural constraints on the properties of KKT points of the margin maximization problem for ReLU nets; this appears in Section A.2. We conclude in Section A.3 by showing how these structural conditions allow for generalization on fresh test data.

## A.1   Training dataset properties

We denote $\mathcal{N}_{\boldsymbol{\theta}}(\mathbf{x}) = \sum_{j \in [m]} v_j \phi(\mathbf{w}_j^\top \mathbf{x} + b_j)$. Thus, $\mathcal{N}_{\boldsymbol{\theta}}$ is a network of width $m$, where the weights in the first layer are $\mathbf{w}_1, \ldots, \mathbf{w}_m$, the bias terms are $b_1, \ldots, b_m$, and the weights in the second layer are $v_1, \ldots, v_m$. We denote $J := [m]$, $J_+ := \{j \in J : v_j > 0\}$, and $J_- := \{j \in J : v_j < 0\}$. Moreover, we denote $I := [n]$, $I_+ := \{i \in I : y_i = 1\}$, and $I_- := \{i \in I : y_i = -1\}$. Finally, we denote $Q := [k]$, $Q_+ := \{q \in Q : y^{(q)} = 1\}$, and $Q_- := \{q \in Q : y^{(q)} = -1\}$.

We denote $p := \max_{q \neq q'} |\langle \boldsymbol{\mu}^{(q)}, \boldsymbol{\mu}^{(q')} \rangle|$. The distribution $\mathcal{D}_{clusters}$ is such that each example $(\mathbf{x}_i, y_i)$ in $\mathcal{S}$ is generated as follows: we draw $q_i \sim \mathcal{U}(Q)$ and $\boldsymbol{\xi}_i \sim \mathsf{N}(\mathbf{0}, \sigma^2 I_d)$ and set $\mathbf{x}_i = \boldsymbol{\mu}^{(q_i)} + \boldsymbol{\xi}_i$ and $y_i = y^{(q_i)}$. We denote $\mathrm{cluster}(i) = q_i$. For $q \in Q$ we denote $I^{(q)} = \{i \in I : \mathrm{cluster}(i) = q\}$. We also denote $\Delta = 4\sigma \sqrt{d} \ln(d)$.

Our goal in this section will be to show that with high probability, the dataset $\mathcal{S}$ satisfies the following properties.

(P1)  For every $i \in I$ we have $\|\boldsymbol{\xi}_i\| \leq \sigma\sqrt{2d}$.

(P2)  For every $i \neq i'$ in $I$ we have $|\langle \boldsymbol{\xi}_i, \boldsymbol{\xi}_{i'} \rangle| \leq \sigma^2\sqrt{2d}\ln(d)$.

(P3)  For every $i \in I$ and $q \in Q$ we have $|\langle \boldsymbol{\mu}^{(q)}, \boldsymbol{\xi}_i \rangle| \leq \sigma\sqrt{d}\ln(d)$.

(P4)  For every $i, i' \in I$ with $\mathrm{cluster}(i) \neq \mathrm{cluster}(i')$ we have $|\langle \mathbf{x}_i, \mathbf{x}_{i'} \rangle| \leq p + \Delta$.

(P5)  For every $i, i' \in I$ with $\mathrm{cluster}(i) = \mathrm{cluster}(i')$ we have $d - \Delta \leq \langle \mathbf{x}_i, \mathbf{x}_{i'} \rangle \leq 3d + \Delta$.

(P6)  For every $q \in Q$ there exists $i \in I$ with $\mathrm{cluster}(i) = q$ (i.e., $I^{(q)} \neq \emptyset$).

More formally, in the remainder of this section we shall show the following proposition.

**Proposition A.1.** *With probability at least* $1 - \left( 3n^2 d^{-\frac{\ln(d)}{2}} + n^2 e^{-d/16} + d^{1-\ln(d)} \right)$, *the dataset* $\mathcal{S}$ *satisfies the properties (P1) through (P6).*

We start with some auxiliary lemmas. The first bounds the norm of $\boldsymbol{\xi}$.

**Lemma A.1.** *Let* $\boldsymbol{\xi} \sim \mathsf{N}(\mathbf{0}, \sigma^2 I_d)$. *Then,*

$$\Pr\left[ \|\boldsymbol{\xi}\| \geq \sigma\sqrt{2d} \right] \leq e^{-d/16} .$$

*Proof.* Note that $\left\| \frac{\boldsymbol{\xi}}{\sigma} \right\|^2$ has the Chi-squared distribution. A concentration bound by Laurent and Massart [LM00, Lemma 1] implies that for all $t > 0$ we have

$$\Pr\left[ \left\| \frac{\boldsymbol{\xi}}{\sigma} \right\|^2 - d \geq 2\sqrt{dt} + 2t \right] \leq e^{-t} .$$

Plugging-in $t = \frac{d}{16}$, we get

$$\Pr\left[ \left\| \frac{\boldsymbol{\xi}}{\sigma} \right\|^2 \geq 2d \right] \leq \Pr\left[ \left\| \frac{\boldsymbol{\xi}}{\sigma} \right\|^2 - d \geq d/2 + d/8 \right] \leq e^{-d/16} .$$

Thus, we have

$$\Pr\left[ \|\boldsymbol{\xi}\| \geq \sigma\sqrt{2d} \right] \leq e^{-d/16} .$$

$\square$

Our next lemma bounds the projection of a Gaussian $\boldsymbol{\xi}'$ onto a fixed vector $\boldsymbol{\xi}$.

**Lemma A.2.** *Let* $\boldsymbol{\xi} \in \mathbb{R}^d$ *and let* $\boldsymbol{\xi}' \sim \mathsf{N}(\mathbf{0}, \sigma^2 I_d)$. *Then,*

$$\Pr\left[ |\langle \boldsymbol{\xi}, \boldsymbol{\xi}' \rangle| \geq \|\boldsymbol{\xi}\|\,\sigma \ln(d) \right] \leq 2d^{-\frac{\ln(d)}{2}} .$$

*Proof.* Note that $\langle \frac{\boldsymbol{\xi}}{\|\boldsymbol{\xi}\|}, \boldsymbol{\xi}' \rangle$ has the distribution $\mathsf{N}(0, \sigma^2)$. By a standard tail bound, we have for every $t \geq 0$ that $\Pr\left[ \left| \langle \frac{\boldsymbol{\xi}}{\|\boldsymbol{\xi}\|}, \boldsymbol{\xi}' \rangle \right| \geq t \right] \leq 2\exp\left( -\frac{t^2}{2\sigma^2} \right)$. Hence,

$$\Pr\left[ \left| \left\langle \frac{\boldsymbol{\xi}}{\|\boldsymbol{\xi}\|}, \boldsymbol{\xi}' \right\rangle \right| \geq \sigma \ln(d) \right] \leq 2\exp\left( -\frac{\sigma^2 \ln^2(d)}{2\sigma^2} \right) = 2d^{-\frac{\ln(d)}{2}} .$$

The lemma now follows immediately. $\square$

We can utilize the two preceding lemmas to show that the pairwise correlations between independent Gaussians is small relative to the norms of the Gaussians.

**Lemma A.3.** *Let* $\boldsymbol{\xi}, \boldsymbol{\xi}'$ *drawn i.i.d. from* $\mathsf{N}(\mathbf{0}, \sigma^2 I_d)$. *Then,*

$$\Pr\left[ |\langle \boldsymbol{\xi}, \boldsymbol{\xi}' \rangle| \geq \sqrt{2d}\ln(d)\sigma^2 \right] \leq e^{-d/16} + 2d^{-\frac{\ln(d)}{2}} .$$

*Proof.* Note that if $|\langle \boldsymbol{\xi}, \boldsymbol{\xi}' \rangle| \geq \sqrt{2d} \ln(d) \sigma^2$ then we have at least one of the following: (1) $\|\boldsymbol{\xi}\| \geq \sigma\sqrt{2d}$; (2) $|\langle \frac{\boldsymbol{\xi}}{\|\boldsymbol{\xi}\|}, \boldsymbol{\xi}' \rangle| \geq \sigma \ln(d)$. We will bound the probability of these events.

First, by Lemma A.1 we have

$$\Pr\left[\|\boldsymbol{\xi}\| \geq \sigma\sqrt{2d}\right] \leq e^{-d/16} \ .$$

Next, by Lemma A.2 we have

$$\Pr\left[\left|\left\langle \frac{\boldsymbol{\xi}}{\|\boldsymbol{\xi}\|}, \boldsymbol{\xi}' \right\rangle\right| \geq \sigma \ln(d)\right] \leq 2d^{-\frac{\ln(d)}{2}} \ .$$

Combining the above displayed equations we conclude that

$$\Pr\left[|\langle \boldsymbol{\xi}, \boldsymbol{\xi}' \rangle| \geq \sqrt{2d} \ln(d) \sigma^2\right] \leq e^{-d/16} + 2d^{-\frac{\ln(d)}{2}} \ .$$

$\square$

Our next lemma bounds the projection of the noise vectors onto any cluster mean.

**Lemma A.4.** *Let $i \in [k]$ and let $\boldsymbol{\xi} \sim \mathsf{N}(\mathbf{0}, \sigma^2 I_d)$. Then*

$$\Pr\left[|\langle \boldsymbol{\mu}^{(i)}, \boldsymbol{\xi} \rangle| \geq \sigma\sqrt{d} \ln(d)\right] \leq 2d^{-\frac{\ln(d)}{2}} \ .$$

*Proof.* Follows immediately from Lemma A.2. $\square$

The following lemmas use bounds on the pairwise interactions between noise vectors and cluster means to bound the correlations between the sums of noises and cluster means.

**Lemma A.5.** *Let $i \neq j$ be indices in $[k]$. Let $\boldsymbol{\xi}, \boldsymbol{\xi}'$ such that the following hold:*

- $|\langle \boldsymbol{\mu}^{(i)}, \boldsymbol{\xi}' \rangle| \leq \sigma\sqrt{d} \ln(d)$.

- $|\langle \boldsymbol{\mu}^{(j)}, \boldsymbol{\xi} \rangle| \leq \sigma\sqrt{d} \ln(d)$.

- $|\langle \boldsymbol{\xi}, \boldsymbol{\xi}' \rangle| \leq \sigma\sqrt{2d} \ln(d)$.

*Then,*

$$|\langle \boldsymbol{\mu}^{(i)} + \boldsymbol{\xi}, \boldsymbol{\mu}^{(j)} + \boldsymbol{\xi}' \rangle| \leq 4\sigma\sqrt{d} \ln(d) + |\langle \boldsymbol{\mu}^{(i)}, \boldsymbol{\mu}^{(j)} \rangle| \ .$$

*Proof.* We have

$$\begin{aligned}
|\langle \boldsymbol{\mu}^{(i)} + \boldsymbol{\xi}, \boldsymbol{\mu}^{(j)} + \boldsymbol{\xi}' \rangle| &\leq |\langle \boldsymbol{\mu}^{(i)}, \boldsymbol{\mu}^{(j)} \rangle| + |\langle \boldsymbol{\mu}^{(i)}, \boldsymbol{\xi}' \rangle| + |\langle \boldsymbol{\xi}, \boldsymbol{\mu}^{(j)} \rangle| + |\langle \boldsymbol{\xi}, \boldsymbol{\xi}' \rangle| \\
&\leq |\langle \boldsymbol{\mu}^{(i)}, \boldsymbol{\mu}^{(j)} \rangle| + \sigma\sqrt{d} \ln(d) + \sigma\sqrt{d} \ln(d) + \sigma\sqrt{2d} \ln(d) \\
&\leq |\langle \boldsymbol{\mu}^{(i)}, \boldsymbol{\mu}^{(j)} \rangle| + 4\sigma\sqrt{d} \ln(d) \ .
\end{aligned}$$

$\square$

**Lemma A.6.** *Let $i \in [k]$, and let $\boldsymbol{\xi}, \boldsymbol{\xi}'$ such that the following hold:*

- $|\langle \boldsymbol{\mu}^{(i)}, \boldsymbol{\xi} \rangle| \leq \sigma\sqrt{d} \ln(d)$.

- $|\langle \boldsymbol{\mu}^{(i)}, \boldsymbol{\xi}' \rangle| \leq \sigma\sqrt{d} \ln(d)$.

- $|\langle \boldsymbol{\xi}, \boldsymbol{\xi}' \rangle| \leq \sigma\sqrt{2d} \ln(d)$.

*Then,*

$$\left|\langle \boldsymbol{\mu}^{(i)} + \boldsymbol{\xi}, \boldsymbol{\mu}^{(i)} + \boldsymbol{\xi}' \rangle - d\right| \leq 4\sigma\sqrt{d} \ln(d) \ .$$

*Proof.* We have

$$\left|\langle \boldsymbol{\mu}^{(i)} + \boldsymbol{\xi}, \boldsymbol{\mu}^{(i)} + \boldsymbol{\xi}'\rangle - d\right| = \left|\langle \boldsymbol{\mu}^{(i)}, \boldsymbol{\xi}'\rangle + \langle \boldsymbol{\xi}, \boldsymbol{\mu}^{(i)}\rangle + \langle \boldsymbol{\xi}, \boldsymbol{\xi}'\rangle\right|$$
$$\leq \sigma\sqrt{d}\ln(d) + \sigma\sqrt{d}\ln(d) + \sigma\sqrt{2d}\ln(d)$$
$$\leq 4\sigma\sqrt{d}\ln(d) \,.$$

$\square$

The next lemma uses bounds on the projection of the noise vector onto cluster means and the norms of the cluster means to derive bounds on the norm of the sum $\boldsymbol{\mu}^{(i)} + \boldsymbol{\xi}$.

**Lemma A.7.** *Let $i \in [k]$, and let $\boldsymbol{\xi}$ such that the following hold:*

- $|\langle \boldsymbol{\mu}^{(i)}, \boldsymbol{\xi}\rangle| \leq \sigma\sqrt{d}\ln(d)$.

- $\|\boldsymbol{\xi}\|^2 \leq 2\sigma^2 d$.

*Then,*

$$d - 2\sigma\sqrt{d}\ln(d) \leq \left\|\boldsymbol{\mu}^{(i)} + \boldsymbol{\xi}\right\|^2 \leq 3d + 2\sigma\sqrt{d}\ln(d) \,.$$

*Proof.* We have

$$\left\|\boldsymbol{\mu}^{(i)} + \boldsymbol{\xi}\right\|^2 = \left\|\boldsymbol{\mu}^{(i)}\right\|^2 + \|\boldsymbol{\xi}\|^2 + 2\langle \boldsymbol{\mu}^{(i)}, \boldsymbol{\xi}\rangle \geq \left\|\boldsymbol{\mu}^{(i)}\right\|^2 - 2\left|\langle \boldsymbol{\mu}^{(i)}, \boldsymbol{\xi}\rangle\right| \geq d - 2\sigma\sqrt{d}\ln(d) \,,$$

and

$$\left\|\boldsymbol{\mu}^{(i)} + \boldsymbol{\xi}\right\|^2 = \left\|\boldsymbol{\mu}^{(i)}\right\|^2 + \|\boldsymbol{\xi}\|^2 + 2\langle \boldsymbol{\mu}^{(i)}, \boldsymbol{\xi}\rangle \leq d + 2\sigma^2 d + 2\sigma\sqrt{d}\ln(d) \leq 3d + 2\sigma\sqrt{d}\ln(d) \,.$$

$\square$

Our final lemma in this section shows that each cluster contains some examples with high probability.

**Lemma A.8.** *With probability at least $1 - d^{1-\ln(d)}$ the dataset $\mathcal{S}$ contains at least one example from each cluster in $[k]$.*

*Proof.* Note that this problem corresponds to the "coupons collector's problem". The probability that $\mathcal{S}$ does not contain points from cluster $j$ is at most

$$\left(1 - \frac{1}{k}\right)^n \leq \exp\left(-\frac{n}{k}\right) \leq \exp\left(-\ln^2(d)\right) = d^{-\ln(d)} \,,$$

where in the second inequality we used $n \geq k\ln^2(d)$. By the union bound, the probability that there is a cluster that does not appear in $\mathcal{S}$ is at most $k \cdot d^{-\ln(d)}$. Since $k \leq d$ this probability is at most $d^{1-\ln(d)}$. $\square$

The proof of Proposition A.1 now follows by putting together Lemmas A.1, A.3, A.4, A.5, A.6, A.7, and A.8, and using $\sigma \leq 1$.

## A.2 Structural implications of the KKT conditions

In this section we show that if the dataset $\mathcal{S}$ satisfies Properties (P1) through (P6), then the KKT conditions impose a number of constraints on the behavior of the neural network. We shall show that these constraints imply that the network will generalize well to unseen test data. The reader may find it useful to refer back to the beginning of Section A.1 before proceeding.

We first outline what types of structural conditions on the KKT points would be useful for understanding generalization. Suppose that $\mathbf{x}$ is a test example coming from cluster $r \in Q_+$. Our hope is

that $\mathcal{N}_{\boldsymbol{\theta}}(\mathbf{x}) > 0$ for such an example. Recall that since $\boldsymbol{\theta}$ satisfies the KKT conditions of Problem (2), then there are $\lambda_1, \ldots, \lambda_n$ such that for every $j \in J$ we have

$$\mathbf{w}_j = \sum_{i \in I} \lambda_i \nabla_{\mathbf{w}_j} \left( y_i \mathcal{N}_{\boldsymbol{\theta}}(\mathbf{x}_i) \right) = \sum_{i \in I} \lambda_i y_i v_j \phi'_{i,j} \mathbf{x}_i , \tag{6}$$

where $\phi'_{i,j}$ is a subgradient of $\phi$ at $\mathbf{w}_j^\top \mathbf{x}_i + b_j$, i.e., if $\mathbf{w}_j^\top \mathbf{x}_i + b_j \neq 0$ then $\phi'_{i,j} = \mathbb{1}[\mathbf{w}_j^\top \mathbf{x}_i + b_j \geq 0]$, and otherwise $\phi'_{i,j}$ is some value in $[0, 1]$. Also we have $\lambda_i \geq 0$ for all $i$, and $\lambda_i = 0$ if $y_i \mathcal{N}_{\boldsymbol{\theta}}(\mathbf{x}_i) \neq 1$. Likewise, we have

$$b_j = \sum_{i \in I} \lambda_i \nabla_{b_j} \left( y_i \mathcal{N}_{\boldsymbol{\theta}}(\mathbf{x}_i) \right) = \sum_{i \in I} \lambda_i y_i v_j \phi'_{i,j} . \tag{7}$$

Now, consider the network output for an input $\mathbf{x}$,

$$\mathcal{N}_{\boldsymbol{\theta}}(\mathbf{x}) = \sum_{j \in J_+} v_j \phi(\mathbf{w}_j^\top \mathbf{x} + b_j) + \sum_{j \in J_-} v_j \phi(\mathbf{w}_j^\top \mathbf{x} + b_j) \geq \sum_{j \in J_+} v_j (\mathbf{w}_j^\top \mathbf{x} + b_j) + \sum_{j \in J_-} v_j \phi(\mathbf{w}_j^\top \mathbf{x} + b_j), \tag{8}$$

where we have used $\phi(z) \geq z$ for all $z \in \mathbb{R}$. This suggests the following possibility: if we can ensure that $\sum_{j \in J_+} v_j(\mathbf{w}_j^\top \mathbf{x} + b_j)$ is large and positive while $\sum_{j \in J_-} v_j \phi(\mathbf{w}_j^\top \mathbf{x} + b_j)$ is not too negative, then the network will accurately classify the example $\mathbf{x}$. Using the KKT conditions and that $r \in Q_+$ (so $y_i = 1$ for $i \in I^{(r)}$), the first term in the above decomposition is equal to

$$\sum_{j \in J_+} v_j(\mathbf{w}_j^\top \mathbf{x} + b_j) = \sum_{j \in J_+} v_j \left[ \sum_{i \in I} \lambda_i y_i v_j \phi'_{i,j}(\mathbf{x}_i^\top \mathbf{x} + 1) \right]$$

$$= \sum_{j \in J_+} \left[ \left( \sum_{i \in I^{(r)}} \lambda_i v_j^2 \phi'_{i,j}(\mathbf{x}_i^\top \mathbf{x} + 1) \right) + \sum_{q \in Q \backslash \{r\}} \sum_{i \in I^{(q)}} \lambda_i y_i v_j^2 \phi'_{i,j}(\mathbf{x}_i^\top \mathbf{x} + 1) \right]$$

$$\geq \left( \sum_{i \in I^{(r)}} \sum_{j \in J_+} \lambda_i v_j^2 \phi'_{i,j}(\mathbf{x}_i^\top \mathbf{x} + 1) \right) - \sum_{q \in Q \backslash \{r\}} \sum_{i \in I^{(q)}} \sum_{j \in J_+} \lambda_i v_j^2 \phi'_{i,j} |\mathbf{x}_i^\top \mathbf{x} + 1| .$$

Since $\mathbf{x}$ comes from cluster $r$ and the clusters are nearly orthogonal, the pairwise correlations $\mathbf{x}_i^\top \mathbf{x}$ will be large and positive when $i \in I^{(r)}$ but will be small in magnitude when $i \in I^{(q)}$ for $q \neq r$. Thus, we can hope that this term will be large and positive if we can show that the quantity $\sum_{i \in I^{(r)}} \sum_{j \in J_+} \lambda_i v_j^2 \phi'_{i,j}$ is not too small relative to the quantity $\sum_{q \in Q \backslash \{r\}} \sum_{i \in I^{(q)}} \sum_{j \in J_+} \lambda_i v_j^2 \phi'_{i,j}$. By similar arguments, in order to show the second term in Eq. (8) is not too negative, we need to understand how the quantity $\sum_{i \in I^{(q)}} \sum_{j \in J_-} \lambda_i v_j^2 \phi'_{i,j}$ varies across different clusters $q \in Q$.

The above sketch motivates a characterization of how the quantities

$$\sum_{i \in I^{(q)}} \sum_{j \in J_+} \lambda_i v_j^2 \phi'_{i,j}, \quad \sum_{i \in I^{(q)}} \sum_{j \in J_-} \lambda_i v_j^2 \phi'_{i,j}$$

relate to each other for different clusters $q \in Q$. We will obtain upper and lower bounds for these quantities in Lemmas A.10 and A.11 below. We now proceed with the proof.

Recall that $\Delta := 4\sigma\sqrt{d}\ln(d)$. Since by Assumption 2.2 we have $k \leq \frac{d - \Delta + 1}{10(p + \Delta + 1)}$, we let $c' \leq \frac{1}{10}$ be such that $k = c' \cdot \frac{d - \Delta + 1}{p + \Delta + 1}$. Note that $d > \Delta$, and more precisely, the following holds

**Lemma A.9.** *We have* $\Delta \leq \frac{d}{21}$.

*Proof.* Recall that $k(p + \Delta + 1) \leq \frac{d - \Delta + 1}{10}$. Since $k \geq 2$ and $p \geq 0$ it implies that $2(\Delta + 1) \leq \frac{d - \Delta + 1}{10}$. Hence, $\Delta \leq \frac{d - 19}{21} \leq \frac{d}{21}$. $\qquad\square$

We now show that the sums of the form $\sum_{i \in I^{(q)}} \sum_{j \in J_\circ} v_j^2 \lambda_i \phi'_{i,j}$, for $\circ \in \{+, -\}$, are never too large.

**Lemma A.10.** *If $\mathcal{S}$ satisfies the properties (P1) through (P6), then for all $q \in Q$ we have*

$$\max\left\{\sum_{i\in I^{(q)}}\sum_{j\in J_+} v_j^2 \lambda_i \phi_{i,j}', \ \sum_{i\in I^{(q)}}\sum_{j\in J_-} v_j^2 \lambda_i \phi_{i,j}'\right\} \leq \frac{1}{(1-2c')(d-\Delta+1)} \ .$$

*Proof.* Let $\alpha_+ = \max_{q\in Q}\left(\sum_{i\in I^{(q)}}\sum_{j\in J_+} v_j^2 \lambda_i \phi_{i,j}'\right)$, and let $\alpha_- = \max_{q\in Q}\left(\sum_{i\in I^{(q)}}\sum_{j\in J_-} v_j^2 \lambda_i \phi_{i,j}'\right)$. Assume w.l.o.g. that $\alpha_+ \geq \alpha_-$ (the proof for the case $\alpha_+ < \alpha_-$ is similar). Let $\alpha = \alpha_+$ and $r \in \mathrm{argmax}_{q\in Q}\left(\sum_{i\in I^{(q)}}\sum_{j\in J_+} v_j^2 \lambda_i \phi_{i,j}'\right)$. Assume towards contradiction that $\alpha > \frac{1}{(1-2c')(d-\Delta+1)}$. Note that we have $\sum_{i\in I^{(r)}} \lambda_i > 0$, since otherwise $\alpha = 0$. Hence, there exists $i' \in I^{(r)}$ with $\lambda_{i'} > 0$, and thus $y_{i'}\mathcal{N}_{\boldsymbol{\theta}}(\mathbf{x}_{i'}) = 1$. By Eq. (6) and (7) for every $j \in J$ we have

$$
\begin{aligned}
\mathbf{w}_j^\top \mathbf{x}_{i'} + b_j &= \sum_{i\in I} \lambda_i y_i v_j \phi_{i,j}' \mathbf{x}_i^\top \mathbf{x}_{i'} + \sum_{i\in I} \lambda_i y_i v_j \phi_{i,j}' \\
&= \sum_{i\in I} \lambda_i y_i v_j \phi_{i,j}' (\mathbf{x}_i^\top \mathbf{x}_{i'} + 1) \\
&= \left(\sum_{q\in Q\setminus\{r\}}\sum_{i\in I^{(q)}} \lambda_i y_i v_j \phi_{i,j}' (\mathbf{x}_i^\top \mathbf{x}_{i'} + 1)\right) + \sum_{i\in I^{(r)}} \lambda_i y_i v_j \phi_{i,j}' (\mathbf{x}_i^\top \mathbf{x}_{i'} + 1) \ . \quad (9)
\end{aligned}
$$

We consider two cases:

**Case 1:** Assume that $r \in Q_+$. We have

$$
\begin{aligned}
1 &= y_{i'}\mathcal{N}_{\boldsymbol{\theta}}(\mathbf{x}_{i'}) \\
&= 1 \cdot \sum_{j\in J} v_j \phi(\mathbf{w}_j^\top \mathbf{x}_{i'} + b_j) \\
&\geq \sum_{j\in J_+} v_j(\mathbf{w}_j^\top \mathbf{x}_{i'} + b_j) + \sum_{j\in J_-} v_j \phi(\mathbf{w}_j^\top \mathbf{x}_{i'} + b_j) \ . \quad (10)
\end{aligned}
$$

By the case assumption $r \in Q_+$, Eq. (9) and our assumptions on the dataset $\mathcal{S}$, we have

$$
\begin{aligned}
\sum_{j\in J_+} v_j(\mathbf{w}_j^\top \mathbf{x}_{i'} + b_j) &= \sum_{j\in J_+}\left[\left(\sum_{q\in Q\setminus\{r\}}\sum_{i\in I^{(q)}} \lambda_i y_i v_j^2 \phi_{i,j}' (\mathbf{x}_i^\top \mathbf{x}_{i'} + 1)\right) + \sum_{i\in I^{(r)}} \lambda_i y_i v_j^2 \phi_{i,j}' (\mathbf{x}_i^\top \mathbf{x}_{i'} + 1)\right] \\
&\geq \sum_{j\in J_+}\left[\left(-\sum_{q\in Q\setminus\{r\}}\sum_{i\in I^{(q)}} \lambda_i v_j^2 \phi_{i,j}' (p+\Delta+1)\right) + \sum_{i\in I^{(r)}} \lambda_i v_j^2 \phi_{i,j}' (d-\Delta+1)\right] \\
&= \left(-(p+\Delta+1)\sum_{q\in Q\setminus\{r\}}\sum_{i\in I^{(q)}}\sum_{j\in J_+} \lambda_i v_j^2 \phi_{i,j}'\right) + (d-\Delta+1)\sum_{i\in I^{(r)}}\sum_{j\in J_+} \lambda_i v_j^2 \phi_{i,j}' \\
&\geq -(p+\Delta+1)k\alpha + (d-\Delta+1)\alpha \ . \quad (11)
\end{aligned}
$$

In the last line we have used the definition $\alpha = \max_{q \in Q}\left(\sum_{i \in I^{(q)}} \sum_{j \in J_+} v_j^2 \lambda_i \phi'_{i,j}\right)$ and that the cluster with index $r$ achieves this maximum. Moreover, using Eq. (9) again we have

$$
\begin{aligned}
\sum_{j \in J_-} v_j \phi(\mathbf{w}_j^\top \mathbf{x}_{i'} + b_j) &= \sum_{j \in J_-} v_j \phi\left(\left(\sum_{q \in Q \setminus \{r\}} \sum_{i \in I^{(q)}} \lambda_i y_i v_j \phi'_{i,j}(\mathbf{x}_i^\top \mathbf{x}_{i'} + 1)\right) + \sum_{i \in I^{(r)}} \lambda_i y_i v_j \phi'_{i,j}(\mathbf{x}_i^\top \mathbf{x}_{i'} + 1)\right) \\
&\geq \sum_{j \in J_-} v_j \phi\left(\left(\sum_{q \in Q \setminus \{r\}} \sum_{i \in I^{(q)}} \lambda_i |v_j| \phi'_{i,j}(p + \Delta + 1)\right) + \sum_{i \in I^{(r)}} \lambda_i v_j \phi'_{i,j}(d - \Delta + 1)\right) \\
&\geq \sum_{j \in J_-} v_j \phi\left(\sum_{q \in Q \setminus \{r\}} \sum_{i \in I^{(q)}} \lambda_i |v_j| \phi'_{i,j}(p + \Delta + 1)\right) \\
&= \sum_{j \in J_-} v_j \left(\sum_{q \in Q \setminus \{r\}} \sum_{i \in I^{(q)}} \lambda_i |v_j| \phi'_{i,j}(p + \Delta + 1)\right) \\
&= -(p + \Delta + 1)\left(\sum_{q \in Q \setminus \{r\}} \sum_{i \in I^{(q)}} \sum_{j \in J_-} \lambda_i v_j^2 \phi'_{i,j}\right) \\
&\geq -(p + \Delta + 1)k\alpha \ .
\end{aligned}
\tag{12}
$$

The first inequality above uses the properties of the dataset $\mathcal{S}$, that $j \in J_-$ and $\phi$ is non-decreasing, as well as the case assumption that $r \in Q_+$. The last inequality uses the definition of $\alpha$. Combining Eq. (10), (11), and (12) we get

$$
\begin{aligned}
1 &\geq -(p + \Delta + 1)k\alpha + (d - \Delta + 1)\alpha - (p + \Delta + 1)k\alpha \\
&= \alpha\left((d - \Delta + 1) - 2k(p + \Delta + 1)\right) \\
&= \alpha\left((d - \Delta + 1) - 2 \cdot \frac{c'(d - \Delta + 1)}{p + \Delta + 1}(p + \Delta + 1)\right) \\
&= \alpha\left((d - \Delta + 1) - 2c'(d - \Delta + 1)\right) \\
&= \alpha(d - \Delta + 1)(1 - 2c') \\
&> \frac{1}{(1 - 2c')(d - \Delta + 1)}(d - \Delta + 1)(1 - 2c') \\
&= 1 \ ,
\end{aligned}
$$

where in the last inequality we used our assumption on $\alpha$. We have thus reached a contradiction following our assumption on $\alpha$ in the case where $r \in Q_+$.

**Case 2:** Assume that $r \in Q_-$. Fix some $j \in J_+$. If $\phi'_{i,j} = 0$ for every $i \in I^{(r)}$ then

$$
\sum_{i \in I^{(r)}} \lambda_i v_j \phi'_{i,j}(d - \Delta + 1) = 0 \leq \sum_{q \in Q \setminus \{r\}} \sum_{i \in I^{(q)}} \lambda_i v_j \phi'_{i,j}(p + \Delta + 1) \ .
\tag{13}
$$

Otherwise, i.e., if there is $s \in I^{(r)}$ such that $\phi'_{s,j} > 0$, then by the definition of $\phi'_{s,j}$ we have $\mathbf{w}_j^\top \mathbf{x}_s + b_j \geq 0$, and hence by Eq. (9) we have

$$
\begin{aligned}
0 &\leq \mathbf{w}_j^\top \mathbf{x}_s + b_j \\
&= \left(\sum_{q \in Q \setminus \{r\}} \sum_{i \in I^{(q)}} \lambda_i y_i v_j \phi'_{i,j}(\mathbf{x}_i^\top \mathbf{x}_s + 1)\right) + \sum_{i \in I^{(r)}} \lambda_i y_i v_j \phi'_{i,j}(\mathbf{x}_i^\top \mathbf{x}_s + 1) \\
&\leq \left(\sum_{q \in Q \setminus \{r\}} \sum_{i \in I^{(q)}} \lambda_i v_j \phi'_{i,j}(p + \Delta + 1)\right) - \sum_{i \in I^{(r)}} \lambda_i v_j \phi'_{i,j}(d - \Delta + 1)
\end{aligned}
$$

Hence, we get again an expression similar to Eq. (13). Thus for any $j \in J_+$ we have,

$$
\sum_{i \in I^{(r)}} \lambda_i v_j \phi'_{i,j} \leq \frac{p + \Delta + 1}{d - \Delta + 1}\left(\sum_{q \in Q \setminus \{r\}} \sum_{i \in I^{(q)}} \lambda_i v_j \phi'_{i,j}\right) \ .
$$

Since this holds for every $j \in J_+$, we get

$$\sum_{i \in I^{(r)}} \sum_{j \in J_+} \lambda_i v_j^2 \phi'_{i,j} = \sum_{j \in J_+} v_j \sum_{i \in I^{(r)}} \lambda_i v_j \phi'_{i,j}$$

$$\leq \sum_{j \in J_+} v_j \cdot \frac{p + \Delta + 1}{d - \Delta + 1} \left( \sum_{q \in Q \setminus \{r\}} \sum_{i \in I^{(q)}} \lambda_i v_j \phi'_{i,j} \right)$$

$$= \frac{p + \Delta + 1}{d - \Delta + 1} \sum_{q \in Q \setminus \{r\}} \sum_{i \in I^{(q)}} \sum_{j \in J_+} \lambda_i v_j^2 \phi'_{i,j}$$

$$\leq \frac{p + \Delta + 1}{d - \Delta + 1} \cdot k \cdot \max_{q \in Q} \left( \sum_{i \in I^{(q)}} \sum_{j \in J_+} \lambda_i v_j^2 \phi'_{i,j} \right)$$

$$= \frac{p + \Delta + 1}{d - \Delta + 1} \cdot \frac{c'(d - \Delta + 1)}{p + \Delta + 1} \cdot \max_{q \in Q} \left( \sum_{i \in I^{(q)}} \sum_{j \in J_+} \lambda_i v_j^2 \phi'_{i,j} \right)$$

$$< \max_{q \in Q} \left( \sum_{i \in I^{(q)}} \sum_{j \in J_+} \lambda_i v_j^2 \phi'_{i,j} \right).$$

Since $r \in \mathrm{argmax}_{q \in Q} \left( \sum_{i \in I^{(q)}} \sum_{j \in J_+} v_j^2 \lambda_i \phi'_{i,j} \right)$, we have reached a contradiction following our assumption on $\alpha$ for the case $r \in Q_-$. This completes the proof that we must have $\alpha \leq \frac{1}{(1-2c')(d-\Delta+1)}$. $\square$

We next show that the relevant sums of the form $\sum_{i \in I^{(q)}} \sum_{j \in J_\circ} v_j^2 \lambda_i \phi'_{i,j}$, for $\circ \in \{+, -\}$, are never too small.

**Lemma A.11.** *If $\mathcal{S}$ satisfies the properties (P1) through (P6), then for all $q \in Q_+$ we have*

$$\sum_{i \in I^{(q)}} \sum_{j \in J_+} v_j^2 \lambda_i \phi'_{i,j} \geq \left( 1 - \frac{c'}{1 - 2c'} \right) \frac{1}{3d + \Delta + 1},$$

*and for all $q \in Q_-$ we have*

$$\sum_{i \in I^{(q)}} \sum_{j \in J_-} v_j^2 \lambda_i \phi'_{i,j} \geq \left( 1 - \frac{c'}{1 - 2c'} \right) \frac{1}{3d + \Delta + 1}.$$

*Proof.* Let $r \in Q_+$ and let $s \in I^{(r)}$. We have

$$1 \leq \mathcal{N}_{\boldsymbol{\theta}}(\mathbf{x}_s) = \sum_{j \in J} v_j \phi(\mathbf{w}_j^\top \mathbf{x}_s + b_j) \leq \sum_{j \in J_+} v_j \phi(\mathbf{w}_j^\top \mathbf{x}_s + b_j) \leq \sum_{j \in J_+} v_j \left| \mathbf{w}_j^\top \mathbf{x}_s + b_j \right|.$$

By Eq. (6) and (7), since $r \in Q_+$ the above equals

$$\sum_{j \in J_+} v_j \left| \sum_{i \in I} \left( \lambda_i y_i v_j \phi'_{i,j} \mathbf{x}_i^\top \mathbf{x}_s + \lambda_i y_i v_j \phi'_{i,j} \right) \right| \leq \sum_{j \in J_+} v_j \sum_{q \in Q} \sum_{i \in I^{(q)}} \left| \lambda_i y_i v_j \phi'_{i,j} (\mathbf{x}_i^\top \mathbf{x}_s + 1) \right|$$

$$= \sum_{j \in J_+} v_j \left[ \left( \sum_{i \in I^{(r)}} \lambda_i v_j \phi'_{i,j} \left| \mathbf{x}_i^\top \mathbf{x}_s + 1 \right| \right) + \sum_{q \in Q \setminus \{r\}} \sum_{i \in I^{(q)}} \lambda_i v_j \phi'_{i,j} \left| \mathbf{x}_i^\top \mathbf{x}_s + 1 \right| \right]$$

$$= \left( \sum_{i \in I^{(r)}} \sum_{j \in J_+} \lambda_i v_j^2 \phi'_{i,j} \left| \mathbf{x}_i^\top \mathbf{x}_s + 1 \right| \right) + \sum_{q \in Q \setminus \{r\}} \sum_{i \in I^{(q)}} \sum_{j \in J_+} \lambda_i v_j^2 \phi'_{i,j} \left| \mathbf{x}_i^\top \mathbf{x}_s + 1 \right|$$

$$\leq \left( (3d + \Delta + 1) \sum_{i \in I^{(r)}} \sum_{j \in J_+} \lambda_i v_j^2 \phi'_{i,j} \right) + (p + \Delta + 1) \sum_{q \in Q \setminus \{r\}} \sum_{i \in I^{(q)}} \sum_{j \in J_+} \lambda_i v_j^2 \phi'_{i,j}$$

The final inequality uses the properties of the dataset $\mathcal{S}$. Combining the above with Lemma A.10 we get

$$1 \leq \left( (3d + \Delta + 1) \sum_{i \in I^{(r)}} \sum_{j \in J_+} \lambda_i v_j^2 \phi'_{i,j} \right) + (p + \Delta + 1)k \cdot \frac{1}{(1 - 2c')(d - \Delta + 1)}$$

$$= \left( (3d + \Delta + 1) \sum_{i \in I^{(r)}} \sum_{j \in J_+} \lambda_i v_j^2 \phi'_{i,j} \right) + (p + \Delta + 1) \cdot \frac{c'(d - \Delta + 1)}{p + \Delta + 1} \cdot \frac{1}{(1 - 2c')(d - \Delta + 1)}$$

$$= \left( (3d + \Delta + 1) \sum_{i \in I^{(r)}} \sum_{j \in J_+} \lambda_i v_j^2 \phi'_{i,j} \right) + \frac{c'}{1 - 2c'} .$$

Therefore,

$$\sum_{i \in I^{(r)}} \sum_{j \in J_+} \lambda_i v_j^2 \phi'_{i,j} \geq \left( 1 - \frac{c'}{1 - 2c'} \right) \frac{1}{3d + \Delta + 1} .$$

By similar arguments with $r \in Q_-$ we also get

$$\sum_{i \in I^{(r)}} \sum_{j \in J_-} \lambda_i v_j^2 \phi'_{i,j} \geq \left( 1 - \frac{c'}{1 - 2c'} \right) \frac{1}{3d + \Delta + 1} .$$

$\square$

Recall that test examples will come from one of the $k$ nearly-orthogonal clusters. Since the clusters are nearly-orthogonal, the pairwise correlations between the test example and training data from the same cluster will be much larger than the pairwise correlations between the test example and training data from the other (nearly-orthogonal) clusters. To characterize the decision boundary of the neural network on test data it therefore suffices to characterize the decision boundary for an example $\mathbf{x}$ that is (1) highly correlated to examples from a given cluster and (2) nearly-orthogonal to samples from other clusters. The next lemma leverages the structural conditions provided in Lemmas A.10 and A.11 to show exactly this.

**Lemma A.12.** *Suppose $\mathcal{S}$ satisfies the properties (P1) through (P6). Let $\mathbf{x} \in \mathbb{R}^d$ and $r \in Q$ be such that for all $i \in I^{(r)}$ we have $\langle \mathbf{x}, \mathbf{x}_i \rangle \in [d - \Delta, d + \Delta]$, and for all $q \in Q \setminus \{r\}$ and $i \in I^{(q)}$ we have $|\langle \mathbf{x}, \mathbf{x}_i \rangle| \leq p + \Delta$. Then, $\mathrm{sign}\left( \mathcal{N}_{\boldsymbol{\theta}}(\mathbf{x}) \right) = y^{(r)}$.*

*Proof.* We prove the claim for $r \in Q_+$. The proof for $r \in Q_-$ is similar. By Eq. (6) and (7), for every $j \in J$ we have

$$\mathbf{w}_j^\top \mathbf{x} + b_j = \left( \sum_{i \in I} \lambda_i y_i v_j \phi'_{i,j} \mathbf{x}_i^\top \mathbf{x} \right) + \sum_{i \in I} \lambda_i y_i v_j \phi'_{i,j}$$

$$= \sum_{i \in I} \lambda_i y_i v_j \phi'_{i,j} (\mathbf{x}_i^\top \mathbf{x} + 1)$$

$$= \left( \sum_{i \in I^{(r)}} \lambda_i v_j \phi'_{i,j} (\mathbf{x}_i^\top \mathbf{x} + 1) \right) + \sum_{q \in Q \setminus \{r\}} \sum_{i \in I^{(q)}} \lambda_i y_i v_j \phi'_{i,j} (\mathbf{x}_i^\top \mathbf{x} + 1) . \quad (14)$$

Now,

$$\mathcal{N}_{\boldsymbol{\theta}}(\mathbf{x}) = \sum_{j \in J} v_j \phi(\mathbf{w}_j^\top \mathbf{x} + b_j) \geq \sum_{j \in J_+} v_j (\mathbf{w}_j^\top \mathbf{x} + b_j) + \sum_{j \in J_-} v_j \phi(\mathbf{w}_j^\top \mathbf{x} + b_j) . \quad (15)$$

By Eq. (14) we have

$$\sum_{j \in J_+} v_j(\mathbf{w}_j^\top \mathbf{x} + b_j) = \sum_{j \in J_+} \left[ \left( \sum_{i \in I^{(r)}} \lambda_i v_j^2 \phi'_{i,j}(\mathbf{x}_i^\top \mathbf{x} + 1) \right) + \sum_{q \in Q \setminus \{r\}} \sum_{i \in I^{(q)}} \lambda_i y_i v_j^2 \phi'_{i,j}(\mathbf{x}_i^\top \mathbf{x} + 1) \right]$$

$$\geq \sum_{j \in J_+} \left[ \left( \sum_{i \in I^{(r)}} \lambda_i v_j^2 \phi'_{i,j}(d - \Delta + 1) \right) - \sum_{q \in Q \setminus \{r\}} \sum_{i \in I^{(q)}} \lambda_i v_j^2 \phi'_{i,j}(p + \Delta + 1) \right]$$

$$= \left( (d - \Delta + 1) \sum_{i \in I^{(r)}} \sum_{j \in J_+} \lambda_i v_j^2 \phi'_{i,j} \right)$$

$$- \left( (p + \Delta + 1) \sum_{q \in Q \setminus \{r\}} \sum_{i \in I^{(q)}} \sum_{j \in J_+} \lambda_i v_j^2 \phi'_{i,j} \right) .$$

By Lemma A.10 and Lemma A.11 the above is at least

$$(d - \Delta + 1) \left( 1 - \frac{c'}{1 - 2c'} \right) \frac{1}{3d + \Delta + 1} - (p + \Delta + 1)k \cdot \frac{1}{(1 - 2c')(d - \Delta + 1)}$$

$$= \left( 1 - \frac{c'}{1 - 2c'} \right) \frac{d - \Delta + 1}{3d + \Delta + 1} - (p + \Delta + 1)c' \cdot \frac{d - \Delta + 1}{p + \Delta + 1} \cdot \frac{1}{(1 - 2c')(d - \Delta + 1)}$$

$$= \left( 1 - \frac{c'}{1 - 2c'} \right) \frac{d - \Delta + 1}{3d + \Delta + 1} - \frac{c'}{1 - 2c'} . \tag{16}$$

Likewise, we have

$$\sum_{j \in J_-} v_j \phi(\mathbf{w}_j^\top \mathbf{x} + b_j) = \sum_{j \in J_-} v_j \phi \left( \left( \sum_{i \in I^{(r)}} \lambda_i v_j \phi'_{i,j}(\mathbf{x}_i^\top \mathbf{x} + 1) \right) + \sum_{q \in Q \setminus \{r\}} \sum_{i \in I^{(q)}} \lambda_i y_i v_j \phi'_{i,j}(\mathbf{x}_i^\top \mathbf{x} + 1) \right)$$

$$\geq \sum_{j \in J_-} v_j \phi \left( \left( \sum_{i \in I^{(r)}} \lambda_i v_j \phi'_{i,j}(d - \Delta + 1) \right) + \sum_{q \in Q \setminus \{r\}} \sum_{i \in I^{(q)}} \lambda_i |v_j| \phi'_{i,j}(p + \Delta + 1) \right)$$

$$\geq \sum_{j \in J_-} v_j \phi \left( \sum_{q \in Q \setminus \{r\}} \sum_{i \in I^{(q)}} \lambda_i |v_j| \phi'_{i,j}(p + \Delta + 1) \right)$$

$$= - \sum_{j \in J_-} \sum_{q \in Q \setminus \{r\}} \sum_{i \in I^{(q)}} \lambda_i v_j^2 \phi'_{i,j}(p + \Delta + 1)$$

$$= -(p + \Delta + 1) \sum_{q \in Q \setminus \{r\}} \sum_{i \in I^{(q)}} \sum_{j \in J_-} \lambda_i v_j^2 \phi'_{i,j} .$$

By Lemma A.10 the above is at least

$$-(p + \Delta + 1)k \cdot \frac{1}{(1 - 2c')(d - \Delta + 1)} = -(p + \Delta + 1)c' \cdot \frac{d - \Delta + 1}{p + \Delta + 1} \cdot \frac{1}{(1 - 2c')(d - \Delta + 1)}$$

$$= -\frac{c'}{1 - 2c'} . \tag{17}$$

Combining Eq. (15), (16), and (17), we get

$$\mathcal{N}_{\boldsymbol{\theta}}(\mathbf{x}) \geq \left( 1 - \frac{c'}{1 - 2c'} \right) \frac{d - \Delta + 1}{3d + \Delta + 1} - \frac{c'}{1 - 2c'} - \frac{c'}{1 - 2c'} .$$

Using $c' \leq \frac{1}{10}$ and $\Delta \leq d$ (which holds by Lemma A.9), the above is at least

$$\frac{7}{8} \cdot \frac{d - \Delta + 1}{3d + \Delta + 1} - \frac{2}{8} \geq \frac{7}{8} \cdot \frac{d - \Delta}{3d + \Delta} - \frac{2}{8} .$$

By Lemma A.9, the displayed equation is at least

$$\frac{7}{8} \cdot \frac{d - d/21}{3d + d/21} - \frac{2}{8} = \frac{7}{8} \cdot \frac{5}{16} - \frac{2}{8} > 0 .$$

$\square$

## A.3  Generalization from KKT conditions

Lemma A.12 shows that in order to show generalization, it suffices to show that with high probability, a test example is highly correlated to one cluster and nearly-orthogonal to all other clusters. In this section we prove that this is the case. We shall re-apply many of the concentration bounds provided in Section A.1 to do so.

**Lemma A.13.** *Suppose $\mathcal{S}$ satisfies Properties (P1) through (P6). Let $r \in Q$ and let $\mathbf{x} = \boldsymbol{\mu}^{(r)} + \boldsymbol{\xi}$ where $\boldsymbol{\xi} \sim \mathsf{N}(\mathbf{0}, \sigma^2 I_d)$. With probability at least $1 - 4nd^{-\frac{\ln(d)}{2}}$ over $\mathbf{x}$ the following hold for all $i \in I$:*

- $|\langle \boldsymbol{\xi}_i, \boldsymbol{\xi} \rangle| \leq \Delta.$

- $|\langle \mathbf{x}_i, \boldsymbol{\xi} \rangle| \leq \Delta.$

- *If $i \notin I^{(r)}$ then $|\langle \mathbf{x}_i, \mathbf{x} \rangle| \leq p + \Delta.$*

- *If $i \in I^{(r)}$ then $|\langle \mathbf{x}_i, \mathbf{x} \rangle - d| \leq \Delta.$*

*Proof.* By our assumption on the dataset $\mathcal{S}$, for all $i \in I$ and $q \in Q$ we have $\|\boldsymbol{\xi}_i\| \leq \sigma\sqrt{2d}$ and $\langle \boldsymbol{\mu}^{(q)}, \boldsymbol{\xi}_i \rangle \leq \sigma\sqrt{d}\ln(d)$. By Lemma A.2 and since $\sigma \leq 1$, for $i \in I$ we have

$$\Pr\left[|\langle \boldsymbol{\xi}, \boldsymbol{\xi}_i \rangle| \geq 2\sigma\sqrt{d}\ln(d)\right] \leq \Pr\left[|\langle \boldsymbol{\xi}, \boldsymbol{\xi}_i \rangle| \geq \sigma\sqrt{2d} \cdot \sigma\ln(d)\right] \leq \Pr\left[|\langle \boldsymbol{\xi}, \boldsymbol{\xi}_i \rangle| \geq \|\boldsymbol{\xi}_i\|\sigma\ln(d)\right] \leq 2d^{-\frac{\ln(d)}{2}} ,$$

and by Lemma A.4 for $q \in Q$ we have

$$\Pr\left[|\langle \boldsymbol{\mu}^{(q)}, \boldsymbol{\xi} \rangle| \geq \sigma\sqrt{d}\ln(d)\right] \leq 2d^{-\frac{\ln(d)}{2}} .$$

Fix some $q \in Q$ and $i \in I^{(q)}$. With probability at least $1 - 4d^{-\frac{\ln(d)}{2}}$ over $\boldsymbol{\xi}$, we have $|\langle \boldsymbol{\xi}, \boldsymbol{\xi}_i \rangle| \leq 2\sigma\sqrt{d}\ln(d)$ and $|\langle \boldsymbol{\mu}^{(q)}, \boldsymbol{\xi} \rangle| \leq \sigma\sqrt{d}\ln(d)$.

Then, the following hold:

- $|\langle \boldsymbol{\xi}_i, \boldsymbol{\xi} \rangle| \leq \Delta.$

- We have
$$|\langle \mathbf{x}_i, \boldsymbol{\xi} \rangle| \leq |\langle \boldsymbol{\mu}^{(q)}, \boldsymbol{\xi} \rangle| + |\langle \boldsymbol{\xi}_i, \boldsymbol{\xi} \rangle| \leq \sigma\sqrt{d}\ln(d) + 2\sigma\sqrt{d}\ln(d) \leq \Delta .$$

- If $q \neq r$, then
$$\begin{aligned}
|\langle \mathbf{x}, \mathbf{x}_i \rangle| &= |\langle \boldsymbol{\mu}^{(r)} + \boldsymbol{\xi}, \boldsymbol{\mu}^{(q)} + \boldsymbol{\xi}_i \rangle| \\
&\leq |\langle \boldsymbol{\mu}^{(r)}, \boldsymbol{\mu}^{(q)} \rangle| + |\langle \boldsymbol{\mu}^{(r)}, \boldsymbol{\xi}_i \rangle| + |\langle \boldsymbol{\xi}, \boldsymbol{\mu}^{(q)} \rangle| + |\langle \boldsymbol{\xi}, \boldsymbol{\xi}_i \rangle| \\
&\leq p + \sigma\sqrt{d}\ln(d) + \sigma\sqrt{d}\ln(d) + 2\sigma\sqrt{d}\ln(d) \\
&= p + \Delta .
\end{aligned}$$

- If $q = r$, then
$$\begin{aligned}
|\langle \mathbf{x}, \mathbf{x}_i \rangle - d| &= \left|\langle \boldsymbol{\mu}^{(r)} + \boldsymbol{\xi}, \boldsymbol{\mu}^{(r)} + \boldsymbol{\xi}_i \rangle - d\right| \\
&= \left|\langle \boldsymbol{\mu}^{(r)}, \boldsymbol{\xi}_i \rangle + \langle \boldsymbol{\xi}, \boldsymbol{\mu}^{(r)} \rangle + \langle \boldsymbol{\xi}, \boldsymbol{\xi}_i \rangle\right| \\
&\leq |\langle \boldsymbol{\mu}^{(r)}, \boldsymbol{\xi}_i \rangle| + |\langle \boldsymbol{\xi}, \boldsymbol{\mu}^{(q)} \rangle| + |\langle \boldsymbol{\xi}, \boldsymbol{\xi}_i \rangle| \\
&\leq \sigma\sqrt{d}\ln(d) + \sigma\sqrt{d}\ln(d) + 2\sigma\sqrt{d}\ln(d) \\
&= \Delta .
\end{aligned}$$

Overall, by the union bound, with probability at least $1 - 4nd^{-\frac{\ln(d)}{2}}$ the requirements hold for all $i \in I$. $\qquad\square$

Theorem A.1 now follows immediately from Proposition A.1 and Lemmas A.12 and A.13.

## B  Proof of Lemma 2.1

Let $\mathbf{x} = \boldsymbol{\mu}^{(j)} + \boldsymbol{\xi}$ where $\boldsymbol{\xi} \sim \mathsf{N}(\mathbf{0}, \sigma^2 I_d)$, and $y = y^{(j)}$. Then,

$$y\mathbf{u}^\top \mathbf{x} = y^{(j)} \sum_{q \in [k]} y^{(q)}(\boldsymbol{\mu}^{(q)})^\top (\boldsymbol{\mu}^{(j)} + \boldsymbol{\xi}) \geq \left\| \boldsymbol{\mu}^{(j)} \right\|^2 - k\left( \max_{q \neq j} |(\boldsymbol{\mu}^{(q)})^\top \boldsymbol{\mu}^{(j)}| + \max_{q \in [k]} |(\boldsymbol{\mu}^{(q)})^\top \boldsymbol{\xi}| \right) .$$

By Lemma A.4 we have $\Pr\left[ |\langle \boldsymbol{\mu}^{(q)}, \boldsymbol{\xi} \rangle| \geq \sigma\sqrt{d}\ln(d) \right] \leq 2d^{-\frac{\ln(d)}{2}}$. Hence, by the union bound with probability at least $1 - 2kd^{-\ln(d)/2} \geq 1 - 2d^{1-\ln(d)/2}$ we have $\max_{q \in [k]} |(\boldsymbol{\mu}^{(q)})^\top \boldsymbol{\xi}| \leq \sigma\sqrt{d}\ln(d)$. Note that by Assumption 2.2 we must have $k \leq d$. Using Assumption 2.2 again, we conclude that with probability at least $1 - 2d^{1-\ln(d)/2}$ we have

$$y\mathbf{u}^\top \mathbf{x} \geq d - k\left( \max_{q \neq j} |(\boldsymbol{\mu}^{(q)})^\top \boldsymbol{\mu}^{(j)}| + \sigma\sqrt{d}\ln(d) \right) \geq d - \frac{d - 4\sigma\sqrt{d}\ln(d) + 1}{10} \geq \frac{9d - 1}{10} > 0 .$$

## C  Proof of Theorem 4.1

We prove the theorem for $r = k$. The proof for $r > k$ follows immediately by adding zero-weight neurons. Consider the network $\mathcal{N}(\mathbf{x}) = \sum_{j=1}^k v_j \phi(\mathbf{w}_j^\top \mathbf{x} + b_j)$ such that for every $j \in [k]$ we have $v_j = y^{(j)}$, $\mathbf{w}_j = \frac{4\boldsymbol{\mu}^{(j)}}{d}$, and $b_j = -2$. Let $q \sim \mathcal{U}([k])$, let $\mathbf{x} = \boldsymbol{\mu}^{(q)} + \boldsymbol{\xi}$ where $\boldsymbol{\xi} \sim \mathsf{N}(\mathbf{0}, \sigma^2 I_d)$, and let $y = y^{(q)}$. Thus $(\mathbf{x}, y)$ is drawn from $\mathcal{D}_{\text{clusters}}$. We have,

$$\mathbf{w}_q^\top \mathbf{x} + b_q = \frac{4(\boldsymbol{\mu}^{(q)})^\top (\boldsymbol{\mu}^{(q)} + \boldsymbol{\xi})}{d} - 2 = \frac{4(d + \langle \boldsymbol{\mu}^{(q)}, \boldsymbol{\xi} \rangle)}{d} - 2 .$$

By Lemma A.4, we have $\Pr\left[ |\langle \boldsymbol{\mu}^{(q)}, \boldsymbol{\xi} \rangle| \geq \sigma\sqrt{d}\ln(d) \right] \leq 2d^{-\frac{\ln(d)}{2}}$. Hence, with probability at least $1 - 2d^{-\frac{\ln(d)}{2}}$ over $\boldsymbol{\xi}$, for large enough $d$ we have

$$\mathbf{w}_q^\top \mathbf{x} + b_q \geq \frac{4(d - \sigma\sqrt{d}\ln(d))}{d} - 2 \geq \frac{4(d - \sqrt{d}\ln(d))}{d} - 2 = 2 - \frac{4\ln(d)}{\sqrt{d}} \geq 1 . \qquad (18)$$

For $j \neq q$ we have

$$\mathbf{w}_j^\top \mathbf{x} + b_j = \frac{4(\boldsymbol{\mu}^{(j)})^\top (\boldsymbol{\mu}^{(q)} + \boldsymbol{\xi})}{d} - 2 \leq \frac{4 \max_{j \neq q} |\langle \boldsymbol{\mu}^{(j)}, \boldsymbol{\mu}^{(q)} \rangle|}{d} + \frac{4(\boldsymbol{\mu}^{(j)})^\top \boldsymbol{\xi}}{d} - 2 ,$$

and thus with probability at least $1 - 2d^{-\frac{\ln(d)}{2}}$ over $\boldsymbol{\xi}$ we have

$$\mathbf{w}_j^\top \mathbf{x} + b_j \leq -2 + \frac{4}{d}\left( \max_{j \neq q} |\langle \boldsymbol{\mu}^{(j)}, \boldsymbol{\mu}^{(q)} \rangle| + \sigma\sqrt{d}\ln(d) \right) .$$

Since by Assumption 2.2 we have $k\left( \max_{j \neq q} |\langle \boldsymbol{\mu}^{(j)}, \boldsymbol{\mu}^{(q)} \rangle| + 4\sigma\sqrt{d}\ln(d) + 1 \right) \leq \frac{d - 4\sigma\sqrt{d}\ln(d) + 1}{10}$, then the displayed equation is at most

$$-2 + \frac{4}{d} \cdot \frac{d+1}{10k} \leq -2 + \frac{4}{d} \cdot \frac{2d}{10} \leq -1 . \qquad (19)$$

Overall, by the union bound, with probability at least $1 - 2kd^{-\frac{\ln(d)}{2}} \geq 1 - 2d^{1-\frac{\ln(d)}{2}} = 1 - o_d(1)$ we have $\mathcal{N}(\mathbf{x}) = \sum_{j=1}^k v_j \phi(\mathbf{w}_j^\top \mathbf{x} + b_j) = v_q(\mathbf{w}_q^\top \mathbf{x} + b_q) + 0$, and

$$\text{sign}(\mathcal{N}(\mathbf{x})) = \text{sign}(v_q) = y^{(q)} = y .$$

We now prove that $\mathcal{N}$ is $\sqrt{d}$-robust w.r.t. $\mathcal{D}_{\mathbf{x}}$. Thus, we show that with probability at least $1 - o_d(1)$ over $\mathbf{x}$, for every $\mathbf{x}' \in \mathbb{R}^d$ such that $\|\mathbf{x} - \mathbf{x}'\| \leq \frac{\sqrt{d}}{8}$ we have $\operatorname{sign}(\mathcal{N}(\mathbf{x})) = \operatorname{sign}(\mathcal{N}(\mathbf{x}'))$. Note that with probability $1 - o_d(1)$, Eq. (18) holds, and Eq. (19) holds for all $j \neq q$, and hence

$$\mathbf{w}_q^\top \mathbf{x}' + b_q = \mathbf{w}_q^\top (\mathbf{x}' - \mathbf{x}) + \left(\mathbf{w}_q^\top \mathbf{x} + b_q\right) \geq -\|\mathbf{w}_q\| \cdot \|\mathbf{x}' - \mathbf{x}\| + 1 \geq -\frac{4}{\sqrt{d}} \cdot \frac{\sqrt{d}}{8} + 1 = \frac{1}{2},$$

and for all $j \neq q$ we have

$$\mathbf{w}_j^\top \mathbf{x}' + b_j = \mathbf{w}_j^\top (\mathbf{x}' - \mathbf{x}) + \left(\mathbf{w}_j^\top \mathbf{x} + b_j\right) \leq \|\mathbf{w}_j\| \cdot \|\mathbf{x}' - \mathbf{x}\| - 1 \leq \frac{4}{\sqrt{d}} \cdot \frac{\sqrt{d}}{8} - 1 = -\frac{1}{2}.$$

Therefore, $\operatorname{sign}(\mathcal{N}(\mathbf{x}')) = \operatorname{sign}(v_q) = y^{(q)} = \operatorname{sign}(\mathcal{N}(\mathbf{x}))$.

# D    Proof of Theorem 4.2

We will prove the following theorem:

**Theorem D.1.** *Let $\mathcal{S} = \{(\mathbf{x}_i, y_i)\}_{i=1}^n \subseteq \mathbb{R}^d \times \{-1, 1\}$ be a training set drawn i.i.d. from the distribution $\mathcal{D}_{clusters}$, where $n \geq k \ln^2(d)$. We denote $Q_+ = \{q \in [k] : y^{(q)} = 1\}$ and $Q_- = \{q \in [k] : y^{(q)} = -1\}$, and assume that $\min\left\{\frac{|Q_+|}{k}, \frac{|Q_-|}{k}\right\} \geq c$ for some $c > 0$. Let $\mathcal{N}_{\boldsymbol{\theta}}$ be a depth-2 ReLU network such that $\boldsymbol{\theta} = [\mathbf{w}_1, \ldots, \mathbf{w}_m, \mathbf{b}, \mathbf{v}]$ is a KKT point of Problem (2). Then, with probability at least*

$$1 - \left(3n^2 d^{-\frac{\ln(d)}{2}} + n^2 e^{-d/16} + d^{1-\ln(d)}\right)$$

*over $\mathcal{S}$, there is a vector $\mathbf{z} = \eta \cdot \sum_{j \in [k]} y^{(j)} \boldsymbol{\mu}^{(j)}$ with $\eta > 0$ and $\|\mathbf{z}\| \leq \mathcal{O}\left(\sqrt{\frac{d}{c^2 k}}\right)$, such that*

$$\Pr_{(\mathbf{x}, y) \sim \mathcal{D}_{clusters}} \left[\operatorname{sign}(\mathcal{N}_{\boldsymbol{\theta}}(\mathbf{x})) \neq \operatorname{sign}(\mathcal{N}_{\boldsymbol{\theta}}(\mathbf{x} - y\mathbf{z}))\right] \geq 1 - 4nd^{-\frac{\ln(d)}{2}}.$$

It is easy to verify that Theorem D.1 implies Theorem 4.2. Indeed, if $\frac{1}{\delta} \leq \frac{1}{3} d^{\ln(d)-1}$ and

$$n \leq \min\left\{\sqrt{\frac{\delta}{3}} \cdot e^{d/32}, \frac{\sqrt{\delta}}{3} \cdot d^{\ln(d)/4}, \frac{\epsilon}{4} \cdot d^{\ln(d)/2}\right\},$$

then we showed in Appendix A that

- $3n^2 d^{-\frac{\ln(d)}{2}} \leq \frac{\delta}{3}$.
- $n^2 e^{-d/16} \leq \frac{\delta}{3}$.
- $d^{1-\ln(d)} \leq \frac{\delta}{3}$.
- $4nd^{-\frac{\ln(d)}{2}} \leq \epsilon$.

Hence, under the above assumptions on $\delta$ and $n$, Theorem D.1 implies that with probability at least $1 - \delta$ over $\mathcal{S}$, we have $\Pr_{(\mathbf{x}, y) \sim \mathcal{D}_{clusters}} \left[\operatorname{sign}(\mathcal{N}_{\boldsymbol{\theta}}(\mathbf{x})) \neq \operatorname{sign}(\mathcal{N}_{\boldsymbol{\theta}}(\mathbf{x} - y\mathbf{z}))\right] \geq 1 - \epsilon$.

We now turn to prove Theorem D.1. The reader may find it useful to refer back to the notations from the proof of Theorem 3.1 in Section A. We shall show that when the dataset $\mathcal{S}$ satisfies the "nice" properties outlined in Properties (P1) through (P6), then every KKT point of Problem (2) is non-robust in the sense stated in the theorem. By Proposition A.1, the dataset $\mathcal{S}$ satisfies these "nice" properties with probability at least $1 - \left(3n^2 d^{-\frac{\ln(d)}{2}} + n^2 e^{-d/16} + d^{1-\ln(d)}\right)$.

In the following lemma, we state several "nice" properties that are satisfied w.h.p. in a test example.

**Lemma D.1.** *Suppose $\mathcal{S}$ satisfies Properties (P1) through (P6). With probability at least $1 - 4nd^{-\frac{\ln(d)}{2}}$ over $\mathbf{x} \sim \mathcal{D}_{\mathbf{x}}$ there exists $r \in Q$ such that the following hold: $\mathbf{x} = \boldsymbol{\mu}^{(r)} + \boldsymbol{\xi}$ where for all $i \in I$ we have $|\langle \mathbf{x}_i, \boldsymbol{\xi} \rangle| \leq \Delta$ and $|\langle \boldsymbol{\xi}_i, \boldsymbol{\xi} \rangle| \leq \Delta$. Also, $|\langle \mathbf{x}_i, \mathbf{x} \rangle - d| \leq \Delta$ for all $i \in I^{(r)}$ and $|\langle \mathbf{x}_i, \mathbf{x} \rangle| \leq p + \Delta$ for all $i \notin I^{(r)}$.*

*Proof.* The lemma follows immediately from Lemma A.13. □

We next show that if the training data and test data are "nice", then KKT points have network outputs that are not too far from the margin.

**Lemma D.2.** *Suppose $\mathcal{S}$ satisfies Properties (P1) through (P6). Let $\mathbf{x} \in \mathbb{R}^d$ and $r \in Q$ such that for all $i \in I^{(r)}$ we have $\langle \mathbf{x}, \mathbf{x}_i \rangle \in [d - \Delta, d + \Delta]$, and for all $q \in Q \setminus \{r\}$ and $i \in I^{(q)}$ we have $|\langle \mathbf{x}, \mathbf{x}_i \rangle| \leq p + \Delta$. Then, $|\mathcal{N}_{\boldsymbol{\theta}}(\mathbf{x})| \leq 2$.*

*Proof.* We prove the claim for $r \in Q_+$. The proof for $r \in Q_-$ is similar. By Lemma A.12 we have $\mathcal{N}_{\boldsymbol{\theta}}(\mathbf{x}) > 0$. We now show that $\mathcal{N}_{\boldsymbol{\theta}}(\mathbf{x}) \leq 2$.

By Eq. (6) and (7), for every $j \in J$ we have

$$
\begin{aligned}
\mathbf{w}_j^\top \mathbf{x} + b_j &= \left( \sum_{i \in I} \lambda_i y_i v_j \phi'_{i,j} \mathbf{x}_i^\top \mathbf{x} \right) + \sum_{i \in I} \lambda_i y_i v_j \phi'_{i,j} \\
&= \sum_{i \in I} \lambda_i y_i v_j \phi'_{i,j} (\mathbf{x}_i^\top \mathbf{x} + 1) \\
&= \left( \sum_{i \in I^{(r)}} \lambda_i v_j \phi'_{i,j} (\mathbf{x}_i^\top \mathbf{x} + 1) \right) + \sum_{q \in Q \setminus \{r\}} \sum_{i \in I^{(q)}} \lambda_i y_i v_j \phi'_{i,j} (\mathbf{x}_i^\top \mathbf{x} + 1) . \quad (20)
\end{aligned}
$$

Now,

$$
\mathcal{N}_{\boldsymbol{\theta}}(\mathbf{x}) = \sum_{j \in J} v_j \phi(\mathbf{w}_j^\top \mathbf{x} + b_j) \leq \sum_{j \in J_+} v_j \left| \mathbf{w}_j^\top \mathbf{x} + b_j \right| .
$$

By Eq. (20) the above is at most

$$
\begin{aligned}
&\sum_{j \in J_+} v_j \left[ \left( \sum_{i \in I^{(r)}} \lambda_i v_j \phi'_{i,j} (d + \Delta + 1) \right) + \sum_{q \in Q \setminus \{r\}} \sum_{i \in I^{(q)}} \lambda_i v_j \phi'_{i,j} (p + \Delta + 1) \right] \\
&= \sum_{j \in J_+} \left[ \left( \sum_{i \in I^{(r)}} \lambda_i v_j^2 \phi'_{i,j} (d + \Delta + 1) \right) + \sum_{q \in Q \setminus \{r\}} \sum_{i \in I^{(q)}} \lambda_i v_j^2 \phi'_{i,j} (p + \Delta + 1) \right] \\
&= \left[ (d + \Delta + 1) \sum_{i \in I^{(r)}} \sum_{j \in J_+} \lambda_i v_j^2 \phi'_{i,j} \right] + \left[ (p + \Delta + 1) \sum_{q \in Q \setminus \{r\}} \sum_{i \in I^{(q)}} \sum_{j \in J_+} \lambda_i v_j^2 \phi'_{i,j} \right] \\
&\leq \left[ (d + \Delta + 1) \cdot \frac{1}{(1 - 2c')(d - \Delta + 1)} \right] + \left[ (p + \Delta + 1)k \cdot \frac{1}{(1 - 2c')(d - \Delta + 1)} \right] ,
\end{aligned}
$$

where in the last inequality we used Lemma A.10. Plugging in $k = c' \cdot \frac{d - \Delta + 1}{p + \Delta + 1}$, the above equals

$$
\begin{aligned}
&\left[ \frac{d + \Delta + 1}{(1 - 2c')(d - \Delta + 1)} \right] + \left[ c' \cdot \frac{d - \Delta + 1}{p + \Delta + 1} \cdot \frac{p + \Delta + 1}{(1 - 2c')(d - \Delta + 1)} \right] \\
&= \frac{d + \Delta + 1}{(1 - 2c')(d - \Delta + 1)} + \frac{c'}{(1 - 2c')} .
\end{aligned}
$$

Using $c' \leq \frac{1}{10}$, the above is at most

$$
\frac{5(d + \Delta + 1)}{4(d - \Delta + 1)} + \frac{1}{8} \leq \frac{5(d + \Delta)}{4(d - \Delta)} + \frac{1}{8} .
$$

By Lemma A.9, the displayed equation is at most

$$
\frac{5(22d/21)}{4(20d/21)} + \frac{1}{8} = \frac{5 \cdot 22}{4 \cdot 20} + \frac{1}{8} = \frac{3}{2} \leq 2 .
$$

□

Next, we show that the inputs to the neurons are not too negative for nice test examples.

**Lemma D.3.** *Suppose $\mathcal{S}$ satisfies Properties (P1) through (P6). Let $r \in Q$ and let $\mathbf{x} = \boldsymbol{\mu}^{(r)} + \boldsymbol{\xi}$ such that for all $i \in I$ we have $|\langle \mathbf{x}_i, \boldsymbol{\xi} \rangle| \leq \Delta$ and $|\langle \boldsymbol{\xi}_i, \boldsymbol{\xi} \rangle| \leq \Delta$. Also, assume that $\langle \mathbf{x}_i, \mathbf{x} \rangle \in [d - \Delta, d + \Delta]$ for all $i \in I^{(r)}$ and $|\langle \mathbf{x}_i, \mathbf{x} \rangle| \leq p + \Delta$ for all $i \notin I^{(r)}$. Then, for all $j \in J$ we have*

$$\mathbf{w}_j^\top \mathbf{x} + b_j \geq -\sum_{i \in I} \lambda_i |v_j| \phi'_{i,j} (2\Delta + p + 1) \,.$$

*Proof.* Suppose towards contradiction that there exists $j \in J$ such that

$$\mathbf{w}_j^\top \mathbf{x} + b_j < -\sum_{i \in I} \lambda_i |v_j| \phi'_{i,j} (2\Delta + p + 1) \leq -\sum_{i \in I} 2\lambda_i |v_j| \phi'_{i,j} \Delta \,. \tag{21}$$

Suppose first that $y^{(r)} v_j < 0$. By Eq. (6) for all $i' \in I^{(r)}$ we have

$$
\begin{aligned}
\mathbf{w}_j^\top \mathbf{x}_{i'} + b_j &= \mathbf{w}_j^\top \mathbf{x} + b_j + \mathbf{w}_j^\top (\mathbf{x}_{i'} - \mathbf{x}) \\
&= \mathbf{w}_j^\top \mathbf{x} + b_j + \mathbf{w}_j^\top (\boldsymbol{\mu}^{(r)} + \boldsymbol{\xi}_{i'} - \boldsymbol{\mu}^{(r)} - \boldsymbol{\xi}) \\
&< -\sum_{i \in I} 2\lambda_i |v_j| \phi'_{i,j} \Delta + \sum_{i \in I} \lambda_i y_i v_j \phi'_{i,j} \mathbf{x}_i^\top (\boldsymbol{\xi}_{i'} - \boldsymbol{\xi}) \\
&= -\sum_{i \in I} 2\lambda_i |v_j| \phi'_{i,j} \Delta - \left( \sum_{i \in I} \lambda_i y_i v_j \phi'_{i,j} \mathbf{x}_i^\top \boldsymbol{\xi} \right) + \left( \sum_{i \in I \setminus \{i'\}} \lambda_i y_i v_j \phi'_{i,j} \mathbf{x}_i^\top \boldsymbol{\xi}_{i'} \right) + \lambda_{i'} y_{i'} v_j \phi'_{i',j} \mathbf{x}_{i'}^\top \boldsymbol{\xi}_{i'} \\
&\leq -\sum_{i \in I} 2\lambda_i |v_j| \phi'_{i,j} \Delta + \left( \sum_{i \in I} \lambda_i |v_j| \phi'_{i,j} \Delta \right) + \left( \sum_{i \in I \setminus \{i'\}} \lambda_i y_i v_j \phi'_{i,j} \mathbf{x}_i^\top \boldsymbol{\xi}_{i'} \right) + \lambda_{i'} y^{(r)} v_j \phi'_{i',j} \mathbf{x}_{i'}^\top \boldsymbol{\xi}_{i'} \,.
\end{aligned}
\tag{22}
$$

Recall that by our assumption on $\mathcal{S}$, for every $i \in I$ and $q \in Q$ we have $|\langle \boldsymbol{\mu}^{(q)}, \boldsymbol{\xi}_i \rangle| \leq \sigma \sqrt{d} \ln(d)$, and for every $s \neq s'$ in $I$ we have $|\langle \boldsymbol{\xi}_s, \boldsymbol{\xi}_{s'} \rangle| \leq \sigma^2 \sqrt{2d} \ln(d)$. Thus, for $q \in Q$ and $i \in I^{(q)}$ such that $i \neq i'$ we have

$$|\mathbf{x}_i^\top \boldsymbol{\xi}_{i'}| \leq |(\boldsymbol{\mu}^{(q)})^\top \boldsymbol{\xi}_{i'}| + |\boldsymbol{\xi}_i^\top \boldsymbol{\xi}_{i'}| \leq \sigma \sqrt{d} \ln(d) + \sigma^2 \sqrt{2d} \ln(d) \leq \Delta \,.$$

Moreover,

$$\mathbf{x}_{i'}^\top \boldsymbol{\xi}_{i'} = (\boldsymbol{\mu}^{(r)})^\top \boldsymbol{\xi}_{i'} + \boldsymbol{\xi}_{i'}^\top \boldsymbol{\xi}_{i'} \geq (\boldsymbol{\mu}^{(r)})^\top \boldsymbol{\xi}_{i'} \geq -\sigma \sqrt{d} \ln(d) \geq -\Delta \,.$$

Using the above displayed equations and the assumption $y^{(r)} v_j < 0$, the RHS in Eq. (22) is at most

$$
\begin{aligned}
&-\sum_{i \in I} 2\lambda_i |v_j| \phi'_{i,j} \Delta + \left( \sum_{i \in I} \lambda_i |v_j| \phi'_{i,j} \Delta \right) + \left( \sum_{i \in I \setminus \{i'\}} \lambda_i |v_j| \phi'_{i,j} \Delta \right) + \lambda_{i'} |v_j| \phi'_{i',j} \Delta \\
&= -\sum_{i \in I} 2\lambda_i |v_j| \phi'_{i,j} \Delta + 2 \left( \sum_{i \in I} \lambda_i |v_j| \phi'_{i,j} \Delta \right) \\
&= 0 \,.
\end{aligned}
$$

Hence, $y^{(r)} v_j < 0$ implies that $\phi'_{i',j} = \mathbb{1}[\mathbf{w}_j^\top \mathbf{x}_{i'} + b_j \geq 0] = 0$ for all $i' \in I^{(r)}$.

By Eq. (6) and (7) we have

$$
\begin{aligned}
\mathbf{w}_j^\top \mathbf{x} + b_j &= \sum_{i \in I} \lambda_i y_i v_j \phi'_{i,j} \mathbf{x}_i^\top \mathbf{x} + \sum_{i \in I} \lambda_i y_i v_j \phi'_{i,j} \\
&= \sum_{i \in I} \lambda_i y_i v_j \phi'_{i,j} (\mathbf{x}_i^\top \mathbf{x} + 1) \\
&= \left[ \sum_{i \in I^{(r)}} \lambda_i y^{(r)} v_j \phi'_{i,j} (\mathbf{x}_i^\top \mathbf{x} + 1) \right] + \left[ \sum_{q \in Q \setminus \{r\}} \sum_{i \in I^{(q)}} \lambda_i y_i v_j \phi'_{i,j} (\mathbf{x}_i^\top \mathbf{x} + 1) \right] \,. \tag{23}
\end{aligned}
$$

We now analyze both terms in the above RHS.

Note that if $y^{(r)}v_j \geq 0$ then

$$\sum_{i \in I^{(r)}} \lambda_i y^{(r)} v_j \phi'_{i,j}(\mathbf{x}_i^\top \mathbf{x} + 1) \geq \sum_{i \in I^{(r)}} \lambda_i y^{(r)} v_j \phi'_{i,j}(d - \Delta + 1) \geq 0 \,,$$

and if $y^{(r)}v_j < 0$ then we have

$$\sum_{i \in I^{(r)}} \lambda_i y^{(r)} v_j \phi'_{i,j}(\mathbf{x}_i^\top \mathbf{x} + 1) = 0$$

since $\phi'_{i',j} = 0$ for all $i' \in I^{(r)}$. Moreover,

$$\sum_{q \in Q \setminus \{r\}} \sum_{i \in I^{(q)}} \lambda_i y_i v_j \phi'_{i,j}(\mathbf{x}_i^\top \mathbf{x} + 1) \geq - \sum_{q \in Q \setminus \{r\}} \sum_{i \in I^{(q)}} \lambda_i |v_j| \phi'_{i,j}(p + \Delta + 1)$$

$$\geq - \sum_{i \in I} \lambda_i |v_j| \phi'_{i,j}(p + \Delta + 1) \,.$$

Plugging the above equations into Eq. (23) we get

$$\mathbf{w}_j^\top \mathbf{x} + b_j \geq - \sum_{i \in I} \lambda_i |v_j| \phi'_{i,j}(p + \Delta + 1) \geq - \sum_{i \in I} \lambda_i |v_j| \phi'_{i,j}(p + 2\Delta + 1) \,,$$

in contradiction to Eq. (21). $\qquad \square$

We now obtain a lower bound for the rate that perturbations in the direction $\mathbf{u} = \sum_{r \in Q} y^{(r)} \boldsymbol{\mu}^{(r)}$ change the inputs to the neurons.

**Lemma D.4.** *Suppose $\mathcal{S}$ satisfies Properties (P1) through (P6). Let $\mathbf{u} = \sum_{r \in Q} y^{(r)} \boldsymbol{\mu}^{(r)}$. For every $j \in J_+$ we have*

$$\mathbf{w}_j^\top \mathbf{u} \geq \sum_{i \in I} \lambda_i v_j \phi'_{i,j} (d - \Delta - kp - k\Delta) \,.$$

*For every $j \in J_-$ we have*

$$\mathbf{w}_j^\top \mathbf{u} \leq \sum_{i \in I} \lambda_i v_j \phi'_{i,j} (d - \Delta - kp - k\Delta) \,.$$

*Proof.* Let $j \in J$. Using Eq. (6) we have

$$\mathbf{w}_j^\top \sum_{r \in Q} y^{(r)} \boldsymbol{\mu}^{(r)} = \sum_{i \in I} \lambda_i y_i v_j \phi'_{i,j} \mathbf{x}_i^\top \sum_{r \in Q} y^{(r)} \boldsymbol{\mu}^{(r)}$$

$$= \sum_{q \in Q} \sum_{i \in I^{(q)}} \lambda_i y^{(q)} v_j \phi'_{i,j} \mathbf{x}_i^\top \left( y^{(q)} \boldsymbol{\mu}^{(q)} + \sum_{r \in Q \setminus \{q\}} y^{(r)} \boldsymbol{\mu}^{(r)} \right)$$

$$= \sum_{q \in Q} \sum_{i \in I^{(q)}} \lambda_i v_j \phi'_{i,j} \left( (y^{(q)})^2 \mathbf{x}_i^\top \boldsymbol{\mu}^{(q)} + \sum_{r \in Q \setminus \{q\}} y^{(q)} y^{(r)} \mathbf{x}_i^\top \boldsymbol{\mu}^{(r)} \right)$$

$$= \sum_{q \in Q} \sum_{i \in I^{(q)}} \lambda_i v_j \phi'_{i,j} \left[ (\boldsymbol{\mu}^{(q)})^\top \boldsymbol{\mu}^{(q)} + \boldsymbol{\xi}_i^\top \boldsymbol{\mu}^{(q)} + \sum_{r \in Q \setminus \{q\}} \left( y^{(q)} y^{(r)} (\boldsymbol{\mu}^{(q)})^\top \boldsymbol{\mu}^{(r)} + y^{(q)} y^{(r)} \boldsymbol{\xi}_i^\top \boldsymbol{\mu}^{(r)} \right) \right] \,.$$

$$(24)$$

For $j \in J_+$ the above is at least

$$\sum_{q \in Q} \sum_{i \in I^{(q)}} \lambda_i v_j \phi'_{i,j} \left( d - \Delta - \sum_{r \in Q \setminus \{q\}} (p + \Delta) \right) \geq \sum_{i \in I} \lambda_i v_j \phi'_{i,j} (d - \Delta - k(p + \Delta)) \,.$$

Similarly, for $j \in J_-$, Eq. (24) is at most

$$\sum_{q \in Q} \sum_{i \in I^{(q)}} \lambda_i v_j \phi'_{i,j} \left( d - \Delta - \sum_{r \in Q \setminus \{q\}} (p + \Delta) \right) \leq \sum_{i \in I} \lambda_i v_j \phi'_{i,j} \left( d - \Delta - k(p + \Delta) \right) .$$

$\square$

We next show that the quantity $d - \Delta - kp - k\Delta$ appearing in the lemma above is strictly positive.

**Lemma D.5.** *We have $d - \Delta - k(p + \Delta) > 0$.*

*Proof.* By Assumption 2.2 and Lemma A.9, we have

$$k(p + \Delta) < k(p + \Delta + 1) \leq \frac{1}{10} \cdot (d - \Delta + 1) = (d - \Delta) - \left[ \frac{9}{10}(d - \Delta) - \frac{1}{10} \right]$$

$$\leq (d - \Delta) - \left[ \frac{9}{10} \cdot \frac{20d}{21} - \frac{1}{10} \right] \leq (d - \Delta) - \left[ \frac{18 \cdot 1}{21} - \frac{1}{10} \right] < d - \Delta .$$

$\square$

Using the above lemmas, we show that a small perturbation to nice inputs suffices for obtaining positive inputs to all hidden neurons.

**Lemma D.6.** *Suppose $\mathcal{S}$ satisfies Properties (P1) through (P6). Let $\mathbf{z} = \eta \sum_{q \in Q} y^{(q)} \boldsymbol{\mu}^{(q)}$ for $\eta \geq \frac{2\Delta + p + 1}{d - \Delta - kp - k\Delta}$. Let $r \in Q$ and let $\mathbf{x} = \boldsymbol{\mu}^{(r)} + \boldsymbol{\xi}$ such that for all $i \in I$ we have $|\langle \mathbf{x}_i, \boldsymbol{\xi} \rangle| \leq \Delta$ and $|\langle \boldsymbol{\xi}_i, \boldsymbol{\xi} \rangle| \leq \Delta$. Also, assume that $\langle \mathbf{x}_i, \mathbf{x} \rangle \in [d - \Delta, d + \Delta]$ for all $i \in I^{(r)}$ and $|\langle \mathbf{x}_i, \mathbf{x} \rangle| \leq p + \Delta$ for all $i \notin I^{(r)}$. Then, we have for all $j \in J_-$ that $\mathbf{w}_j^\top (\mathbf{x} - \mathbf{z}) + b_j \geq 0$, and for all $j \in J_+$ that $\mathbf{w}_j^\top (\mathbf{x} + \mathbf{z}) + b_j \geq 0$.*

*Proof.* Let $j \in J_-$. By Lemma D.3, we have

$$\mathbf{w}_j^\top \mathbf{x} + b_j \geq \sum_{i \in I} \lambda_i v_j \phi'_{i,j} (2\Delta + p + 1) .$$

By Lemma D.4, we have

$$-\mathbf{w}_j^\top \mathbf{z} \geq -\eta \sum_{i \in I} \lambda_i v_j \phi'_{i,j} (d - \Delta - kp - k\Delta) .$$

Combining the last two displayed equations, we get

$$\mathbf{w}_j^\top (\mathbf{x} - \mathbf{z}) + b_j = \mathbf{w}_j^\top \mathbf{x} + b_j - \mathbf{w}_j^\top \mathbf{z}$$

$$\geq \sum_{i \in I} \lambda_i v_j \phi'_{i,j} (2\Delta + p + 1) - \eta \sum_{i \in I} \lambda_i v_j \phi'_{i,j} (d - \Delta - kp - k\Delta)$$

$$= \sum_{i \in I} \lambda_i v_j \phi'_{i,j} (2\Delta + p + 1 - \eta(d - \Delta - kp - k\Delta)) .$$

Note that by Lemma D.5 we have $d - \Delta - kp - k\Delta > 0$. Hence, for $\eta \geq \frac{2\Delta + p + 1}{d - \Delta - kp - k\Delta}$ we have $\mathbf{w}_j^\top (\mathbf{x} - \mathbf{z}) + b_j \geq 0$.

The proof for $j \in J_+$ is similar. Namely, by Lemmas D.3 and D.4, we have

$$\mathbf{w}_j^\top (\mathbf{x} + \mathbf{z}) + b_j = \mathbf{w}_j^\top \mathbf{x} + b_j + \mathbf{w}_j^\top \mathbf{z}$$

$$\geq -\sum_{i \in I} \lambda_i v_j \phi'_{i,j} (2\Delta + p + 1) + \eta \sum_{i \in I} \lambda_i v_j \phi'_{i,j} (d - \Delta - kp - k\Delta)$$

$$= \sum_{i \in I} \lambda_i v_j \phi'_{i,j} (-2\Delta - p - 1 + \eta(d - \Delta - kp - k\Delta)) ,$$

and hence for $\eta \geq \frac{2\Delta + p + 1}{d - \Delta - kp - k\Delta}$ we get $\mathbf{w}_j^\top (\mathbf{x} + \mathbf{z}) + b_j \geq 0$.

$\square$

Let now $\eta_1 = \frac{2\Delta + p + 1}{d - \Delta - kp - k\Delta}$ and $\eta_2 = \frac{3(3d + \Delta + 1)(1 - 2c')}{(d - \Delta - kp - k\Delta)(1 - 3c')ck}$. Note that by Lemma D.5 both $\eta_1$ and $\eta_2$ are positive. We denote $\mathbf{z} = (\eta_1 + \eta_2) \sum_{q \in Q} y^{(q)} \boldsymbol{\mu}^{(q)}$.

In the next lemma, we show that perturbing nice test examples from positive clusters with the vector $-\mathbf{z}$ changes the sign of the network output.

**Lemma D.7.** *Suppose $\mathcal{S}$ satisfies Properties (P1) through (P6). Let $r \in Q_+$ and let $\mathbf{x} = \boldsymbol{\mu}^{(r)} + \boldsymbol{\xi}$ such that for all $i \in I$ we have $|\langle \mathbf{x}_i, \boldsymbol{\xi} \rangle| \leq \Delta$ and $|\langle \boldsymbol{\xi}_i, \boldsymbol{\xi} \rangle| \leq \Delta$. Also, assume that $\langle \mathbf{x}_i, \mathbf{x} \rangle \in [d - \Delta, d + \Delta]$ for all $i \in I^{(r)}$ and $|\langle \mathbf{x}_i, \mathbf{x} \rangle| \leq p + \Delta$ for all $i \notin I^{(r)}$. Then, $\mathcal{N}_{\boldsymbol{\theta}}(\mathbf{x} - \mathbf{z}) \leq -1$.*

*Proof.* We denote $\mathbf{x}' = \mathbf{x} - \mathbf{z}$. By Lemma D.4, for every $j \in J_+$ we have

$$\mathbf{w}_j^\top \mathbf{x}' + b_j = \mathbf{w}_j^\top \mathbf{x} + b_j - \mathbf{w}_j^\top (\eta_1 + \eta_2) \sum_{q \in Q} y^{(q)} \boldsymbol{\mu}^{(q)}$$

$$\leq \mathbf{w}_j^\top \mathbf{x} + b_j - (\eta_1 + \eta_2) \sum_{i \in I} \lambda_i v_j \phi'_{i,j} (d - \Delta - kp - k\Delta) \ .$$

By Lemma D.5 we get

$$\mathbf{w}_j^\top \mathbf{x}' + b_j \leq \mathbf{w}_j^\top \mathbf{x} + b_j. \tag{25}$$

Thus, in the neurons $J_+$ the input does not increase when moving from $\mathbf{x}$ to $\mathbf{x}'$.

Consider now $j \in J_-$. Let $\tilde{\mathbf{x}} = \mathbf{x} - \eta_1 \sum_{q \in Q} y^{(q)} \boldsymbol{\mu}^{(q)}$. By Lemma D.6, we have $\mathbf{w}_j^\top \tilde{\mathbf{x}} + b_j \geq 0$. Also, by Lemma D.4, we have

$$\mathbf{w}_j^\top \tilde{\mathbf{x}} + b_j = \mathbf{w}_j^\top \mathbf{x} + b_j - \mathbf{w}_j^\top \eta_1 \sum_{q \in Q} y^{(q)} \boldsymbol{\mu}^{(q)}$$

$$\geq \mathbf{w}_j^\top \mathbf{x} + b_j - \eta_1 \sum_{i \in I} \lambda_i v_j \phi'_{i,j} (d - \Delta - kp - k\Delta) \ ,$$

and by Lemma D.5 the above is at least $\mathbf{w}_j^\top \mathbf{x} + b_j$. Thus, when moving from $\mathbf{x}$ to $\tilde{\mathbf{x}}$ the input to the neurons $J_-$ can only increase, and at $\tilde{\mathbf{x}}$ it is non-negative.

Next, we move from $\tilde{\mathbf{x}}$ to $\mathbf{x}'$. We have

$$\mathbf{w}_j^\top \mathbf{x}' + b_j = \mathbf{w}_j^\top \tilde{\mathbf{x}} + b_j - \eta_2 \mathbf{w}_j^\top \sum_{q \in Q} y^{(q)} \boldsymbol{\mu}^{(q)} \geq \max \left\{ 0, \mathbf{w}_j^\top \mathbf{x} + b_j \right\} - \eta_2 \mathbf{w}_j^\top \sum_{q \in Q} y^{(q)} \boldsymbol{\mu}^{(q)} \ .$$

By Lemma D.4, the above is at least

$$\max \left\{ 0, \mathbf{w}_j^\top \mathbf{x} + b_j \right\} - \eta_2 \sum_{i \in I} \lambda_i v_j \phi'_{i,j} (d - \Delta - kp - k\Delta) \geq 0 \ , \tag{26}$$

where in the last inequality we use Lemma D.5.

Overall, we have

$$\mathcal{N}_{\boldsymbol{\theta}}(\mathbf{x}') = \left[ \sum_{j \in J_+} v_j \phi(\mathbf{w}_j^\top \mathbf{x}' + b_j) \right] + \left[ \sum_{j \in J_-} v_j \phi(\mathbf{w}_j^\top \mathbf{x}' + b_j) \right]$$

$$\overset{(i)}{=} \left[ \sum_{j \in J_+} v_j \phi(\mathbf{w}_j^\top \mathbf{x}' + b_j) \right] + \left[ \sum_{j \in J_-} v_j (\mathbf{w}_j^\top \mathbf{x}' + b_j) \right]$$

$$\overset{(ii)}{\leq} \left[ \sum_{j \in J_+} v_j \phi(\mathbf{w}_j^\top \mathbf{x} + b_j) \right] +$$

$$\left[ \sum_{j \in J_-} v_j \left( \max \left\{ 0, \mathbf{w}_j^\top \mathbf{x} + b_j \right\} - \eta_2 \sum_{i \in I} \lambda_i v_j \phi'_{i,j} (d - \Delta - kp - k\Delta) \right) \right] \ ,$$

where in $(i)$ we used Eq. (26), and in $(ii)$ we used both Eq. (25) and Eq. (26). Now, the above equals

$$\left[ \sum_{j \in J} v_j \phi(\mathbf{w}_j^\top \mathbf{x} + b_j) \right] - \left[ \sum_{j \in J_-} v_j \eta_2 \sum_{i \in I} \lambda_i v_j \phi'_{i,j} \left( d - \Delta - kp - k\Delta \right) \right]$$

$$= \mathcal{N}_{\boldsymbol{\theta}}(\mathbf{x}) - \eta_2 \left( d - \Delta - kp - k\Delta \right) \left[ \sum_{q' \in Q} \sum_{i \in I(q')} \sum_{j \in J_-} \lambda_i v_j^2 \phi'_{i,j} \right] .$$

Combining the above with Lemma D.2, Lemma A.11, and Lemma D.5, we get

$$\mathcal{N}_{\boldsymbol{\theta}}(\mathbf{x}') \le 2 - \eta_2 \left( d - \Delta - kp - k\Delta \right) \left[ \sum_{q' \in Q_-} \sum_{i \in I(q')} \sum_{j \in J_-} \lambda_i v_j^2 \phi'_{i,j} \right]$$

$$\le 2 - \eta_2 \left( d - \Delta - kp - k\Delta \right) |Q_-| \left( 1 - \frac{c'}{1 - 2c'} \right) \frac{1}{3d + \Delta + 1}$$

$$\le 2 - \eta_2 \left( d - \Delta - kp - k\Delta \right) ck \left( \frac{1 - 3c'}{1 - 2c'} \right) \frac{1}{3d + \Delta + 1} .$$

For

$$\eta_2 = \frac{3(3d + \Delta + 1)(1 - 2c')}{(d - \Delta - kp - k\Delta)(1 - 3c')ck}$$

we conclude that $\mathcal{N}_{\boldsymbol{\theta}}(\mathbf{x}')$ is at most $-1$. $\qquad \square$

Next, we show that perturbing nice test examples from negative clusters with the vector $\mathbf{z}$ changes the sign of the network output.

**Lemma D.8.** *Suppose $\mathcal{S}$ satisfies Properties (P1) through (P6). Let $r \in Q_-$ and let $\mathbf{x} = \boldsymbol{\mu}^{(r)} + \boldsymbol{\xi}$ such that for all $i \in I$ we have $|\langle \mathbf{x}_i, \boldsymbol{\xi} \rangle| \le \Delta$ and $|\langle \boldsymbol{\xi}_i, \boldsymbol{\xi} \rangle| \le \Delta$. Also, assume that $\langle \mathbf{x}_i, \mathbf{x} \rangle \in [d - \Delta, d + \Delta]$ for all $i \in I^{(r)}$ and $|\langle \mathbf{x}_i, \mathbf{x} \rangle| \le p + \Delta$ for all $i \notin I^{(r)}$. Then, $\mathcal{N}_{\boldsymbol{\theta}}(\mathbf{x} + \mathbf{z}) \ge 1$.*

*Proof.* The proof follows similar arguments to the proof of Lemma D.7. We provide it here for completeness.

We denote $\mathbf{x}' = \mathbf{x} + \mathbf{z}$. By Lemma D.4, for every $j \in J_-$ we have

$$\mathbf{w}_j^\top \mathbf{x}' + b_j = \mathbf{w}_j^\top \mathbf{x} + b_j + \mathbf{w}_j^\top (\eta_1 + \eta_2) \sum_{q \in Q} y^{(q)} \boldsymbol{\mu}^{(q)}$$

$$\le \mathbf{w}_j^\top \mathbf{x} + b_j + (\eta_1 + \eta_2) \sum_{i \in I} \lambda_i v_j \phi'_{i,j} \left( d - \Delta - kp - k\Delta \right) .$$

By Lemma D.5 we get

$$\mathbf{w}_j^\top \mathbf{x}' + b_j \le \mathbf{w}_j^\top \mathbf{x} + b_j. \tag{27}$$

Thus, in the neurons $J_-$ the input does not increase when moving from $\mathbf{x}$ to $\mathbf{x}'$.

Consider now $j \in J_+$. Let $\tilde{\mathbf{x}} = \mathbf{x} + \eta_1 \sum_{q \in Q} y^{(q)} \boldsymbol{\mu}^{(q)}$. By Lemma D.6, we have $\mathbf{w}_j^\top \tilde{\mathbf{x}} + b_j \ge 0$. Also, by Lemma D.4, we have

$$\mathbf{w}_j^\top \tilde{\mathbf{x}} + b_j = \mathbf{w}_j^\top \mathbf{x} + b_j + \mathbf{w}_j^\top \eta_1 \sum_{q \in Q} y^{(q)} \boldsymbol{\mu}^{(q)}$$

$$\ge \mathbf{w}_j^\top \mathbf{x} + b_j + \eta_1 \sum_{i \in I} \lambda_i v_j \phi'_{i,j} \left( d - \Delta - kp - k\Delta \right) ,$$

and by Lemma D.5 the above is at least $\mathbf{w}_j^\top \mathbf{x} + b_j$. Thus, when moving from $\mathbf{x}$ to $\tilde{\mathbf{x}}$ the input to the neurons $J_+$ can only increase, and at $\tilde{\mathbf{x}}$ it is non-negative.

Next, we move from $\tilde{\mathbf{x}}$ to $\mathbf{x}'$. We have

$$\mathbf{w}_j^\top \mathbf{x}' + b_j = \mathbf{w}_j^\top \tilde{\mathbf{x}} + b_j + \eta_2 \mathbf{w}_j^\top \sum_{q \in Q} y^{(q)} \boldsymbol{\mu}^{(q)} \ge \max \left\{ 0, \mathbf{w}_j^\top \mathbf{x} + b_j \right\} + \eta_2 \mathbf{w}_j^\top \sum_{q \in Q} y^{(q)} \boldsymbol{\mu}^{(q)} .$$

By Lemma D.4, the above is at least

$$\max\left\{0, \mathbf{w}_j^\top \mathbf{x} + b_j\right\} + \eta_2 \sum_{i \in I} \lambda_i v_j \phi'_{i,j} \left(d - \Delta - kp - k\Delta\right) \geq 0 \,, \tag{28}$$

where in the last inequality we use Lemma D.5.

Overall, we have

$$
\begin{aligned}
\mathcal{N}_{\boldsymbol{\theta}}(\mathbf{x}') &= \left[\sum_{j \in J_-} v_j \phi(\mathbf{w}_j^\top \mathbf{x}' + b_j)\right] + \left[\sum_{j \in J_+} v_j \phi(\mathbf{w}_j^\top \mathbf{x}' + b_j)\right] \\
&\overset{(i)}{=} \left[\sum_{j \in J_-} v_j \phi(\mathbf{w}_j^\top \mathbf{x}' + b_j)\right] + \left[\sum_{j \in J_+} v_j (\mathbf{w}_j^\top \mathbf{x}' + b_j)\right] \\
&\overset{(ii)}{\geq} \left[\sum_{j \in J_-} v_j \phi(\mathbf{w}_j^\top \mathbf{x} + b_j)\right] + \\
&\qquad \left[\sum_{j \in J_+} v_j \left(\max\left\{0, \mathbf{w}_j^\top \mathbf{x} + b_j\right\} + \eta_2 \sum_{i \in I} \lambda_i v_j \phi'_{i,j} \left(d - \Delta - kp - k\Delta\right)\right)\right] \,,
\end{aligned}
$$

where in $(i)$ we used Eq. (28), and in $(ii)$ we used both Eq. (27) and Eq. (28). Now, the above equals

$$
\begin{aligned}
&\left[\sum_{j \in J} v_j \phi(\mathbf{w}_j^\top \mathbf{x} + b_j)\right] + \left[\sum_{j \in J_+} v_j \eta_2 \sum_{i \in I} \lambda_i v_j \phi'_{i,j} \left(d - \Delta - kp - k\Delta\right)\right] \\
&= \mathcal{N}_{\boldsymbol{\theta}}(\mathbf{x}) + \eta_2 \left(d - \Delta - kp - k\Delta\right) \left[\sum_{q' \in Q} \sum_{i \in I^{(q')}} \sum_{j \in J_+} \lambda_i v_j^2 \phi'_{i,j}\right] \,.
\end{aligned}
$$

Combining the above with Lemma D.2, Lemma A.11, and Lemma D.5, we get

$$
\begin{aligned}
\mathcal{N}_{\boldsymbol{\theta}}(\mathbf{x}') &\geq -2 + \eta_2 \left(d - \Delta - kp - k\Delta\right) \left[\sum_{q' \in Q_+} \sum_{i \in I^{(q')}} \sum_{j \in J_+} \lambda_i v_j^2 \phi'_{i,j}\right] \\
&\geq -2 + \eta_2 \left(d - \Delta - kp - k\Delta\right) |Q_+| \left(1 - \frac{c'}{1 - 2c'}\right) \frac{1}{3d + \Delta + 1} \\
&\geq -2 + \eta_2 \left(d - \Delta - kp - k\Delta\right) ck \left(\frac{1 - 3c'}{1 - 2c'}\right) \frac{1}{3d + \Delta + 1} \,.
\end{aligned}
$$

Plugging-in $\eta_2$, we conclude that $\mathcal{N}_{\boldsymbol{\theta}}(\mathbf{x}')$ is at least 1. $\qquad\square$

Finally, we show that the scale of the perturbation $\mathbf{z}$ is small when $k$ is large.

**Lemma D.9.** *We have* $\|\mathbf{z}\| = \mathcal{O}\left(\sqrt{\frac{d}{c^2 k}}\right)$.

*Proof.* We have

$$\|\mathbf{z}\|^2 = (\eta_1 + \eta_2)^2 \left\|\sum_{q \in Q} y^{(q)} \boldsymbol{\mu}^{(q)}\right\|^2$$

Now,

$$
\begin{aligned}
(\eta_1 + \eta_2)^2 &= \left( \frac{2\Delta + p + 1}{d - \Delta - kp - k\Delta} + \frac{3(3d + \Delta + 1)(1 - 2c')}{(d - \Delta - kp - k\Delta)(1 - 3c')ck} \right)^2 \\
&\leq \left( \frac{2(p + \Delta + 1)}{d - \Delta - k(p + \Delta + 1)} + \frac{3(3d + \Delta + 1)(1 - 2c')}{(d - \Delta - k(p + \Delta + 1))(1 - 3c')ck} \right)^2 \\
&\overset{(i)}{=} \left( \frac{2c'(d - \Delta + 1)/k}{d - \Delta - c'(d - \Delta + 1)} + \frac{3(3d + \Delta + 1)(1 - 2c')}{(d - \Delta - c'(d - \Delta + 1))(1 - 3c')ck} \right)^2 \\
&\overset{(ii)}{\leq} \left( \frac{1}{k} \cdot \frac{\frac{2}{10}(d + 1)}{d - \frac{d}{21} - \frac{1}{10}(d + 1)} + \frac{1}{ck} \cdot \frac{3(3d + \frac{d}{21} + 1)(1 - \frac{2}{10})}{(d - \frac{d}{21} - \frac{1}{10}(d + 1))(1 - \frac{3}{10})} \right)^2 \\
&\leq \left( \frac{1}{k} \cdot \frac{\mathcal{O}(d)}{\Omega(d)} + \frac{1}{ck} \cdot \frac{\mathcal{O}(d)}{\Omega(d)} \right)^2 \\
&\leq \mathcal{O} \left( \frac{1}{c^2 k^2} \right),
\end{aligned}
$$

where in $(i)$ we used $k = c' \cdot \frac{d - \Delta + 1}{p + \Delta + 1}$, and in $(ii)$ we used Lemma A.9.

Moreover,

$$
\begin{aligned}
\left\| \sum_{q \in Q} y^{(q)} \boldsymbol{\mu}^{(q)} \right\|^2 &= \sum_{r \in Q} \sum_{q \in Q} y^{(r)} y^{(q)} \langle \boldsymbol{\mu}^{(r)}, \boldsymbol{\mu}^{(q)} \rangle \\
&= \sum_{r \in Q} \left[ \left\| \boldsymbol{\mu}^{(r)} \right\|^2 + \sum_{q \neq r} y^{(r)} y^{(q)} \langle \boldsymbol{\mu}^{(r)}, \boldsymbol{\mu}^{(q)} \rangle \right] \\
&\leq kd + k^2 p \\
&= kd + k^2 \left( \frac{c'(d - \Delta + 1)}{k} - \Delta - 1 \right) \\
&\leq kd + kc'(d - \Delta + 1) \\
&\leq \mathcal{O}(kd).
\end{aligned}
$$

Overall, we get

$$
\|\mathbf{z}\|^2 \leq \mathcal{O} \left( \frac{d}{c^2 k} \right).
$$

$\square$

The theorem now follows immediately from Lemmas D.1, D.7, D.8, and D.9.