# OpenReview forum: "The Double-Edged Sword of Implicit Bias: Generalization vs. Robustness in ReLU Networks"
_NeurIPS.cc/2023/Conference — NeurIPS 2023 poster_

### Official Review · Reviewer_rYUh · 2023-07-02

**Soundness:** 3 good
**Presentation:** 3 good
**Contribution:** 3 good
**Rating:** 7
**Confidence:** 4

**Summary:**

This paper studies the implicit bias of gradient descent for two-layer ReLU networks for cluster data distribution, and shows the implicit bias towards solutions that generalize well but are vulnerable to adversarial examples.


**Strengths:**

The paper builds upon previous work by Vardi et al, removing the orthogonality assumption of the training data, and considering the robustness of test data instead of training data. Although the analysis is via understanding the KKT points, leveraging the previous work result that the implicit bias of gradient low for homogeneous NN given over exp or logistic loss, such regime belongs to the rich regime instead of the lazy regime, which should be more interesting to understand. Therefore I believe the paper provides good contributions to understanding the generalization and robustness of two-layer NN via gradient methods.

Moreover, the paper provides a complementary result of Bubeck and Sellke, showing that even if the network is overparameterized, the implicit bias of gradient flow prevents convergence to robust solutions. This paper gives concrete examples in which the network generalize well but is non-robust, and the perturbation is independent of the network width. Note that Bubeck and Sellke paper consider regression setting whereas this paper consider classification setting.

**Weaknesses:**

Instead of the orthogonality of the training samples as assumed by Vardi et al, this paper turns out to have a clustered assumption on data distribution. I understand such an assumption comes from the proof technique, but it still restricts the setting too much that cannot capture the real scenario.

**Questions:**

1. I'm wondering whether assumption 2.2 (3) can be further simplified by a relationship between k and d, currently the LHS has both d and k.

---

> ### Author Rebuttal · Authors · 2023-08-08
>
>
> We thank the reviewer for the positive feedback.
>
> Regarding the reviewer’s question: Assumption 2.2 (3) essentially requires that $k$ cannot be too large and the correlations between cluster means cannot be too large. We will discuss this inequality in the camera-ready version to make it a bit easier to digest.

---

### Official Review · Reviewer_ZRWm · 2023-07-04

**Soundness:** 4 excellent
**Presentation:** 3 good
**Contribution:** 2 fair
**Rating:** 5
**Confidence:** 2

**Summary:**

In this paper the authors study the implicit bias of two layers neural networks with ReLU activation, in the setting where the data is composed of independant clusters that "dont' overlap" (i.e the probability of having the point of a cluster falling outo the support of another is small, e.g Gaussian of subgaussian). They study both the generalization properties and robustness of the solution obtained by training such network on those data.

They show that if the network suceed to fit the data (i.e the train loss falls below some threshold) and if we follow the gradient flow:
* then the weights converge *in direction*
* the network generalizes well
* however it converges to a non robust solution (if the dimension is high, the number of clusters is small, and the number of points is small too)
* whereas in the same setting robust solutions exist! But they are not "found" by following the gradient flow
* the adversarial attack associated is universal and transferable


**Strengths:**

The paper does a good job at explaining the technical hypothesis, their significance, and the practical consequences.

The "proof idea" sections contain the right level of details to grasp the ideas.

The paper is easy to read despite its technical content.

While I am not familiar with the topic, the literature review seems thorough, in particular the contributions of the paper compared to existing works are clear.

The main conclusion is significant in my opinion: "However, the implicit bias also leads to non-robust solutions (susceptible to small adversarial 2-perturbations), even though robust networks that fit the data exist.", with the additionnal properties that "the adversarial attack associated is universal and transferable".

Definition 4.1 seems to be new, and it is an interesting concept to define robustness.

**Weaknesses:**

The main weakness of the contribution lies in the hypothesis, that are extremely restrictive, and sometimes even more than it seems at first glance.

It is not clear if those hypothesis have a chance to hold on high dimensional image space, and if the main steps of the reasoning can be adapted to this setting, or if this is just specific to this particular setting and problem (see the section "Limitations" below about the setting).

## Thm 4.2

In Thm 4.2 the quantity is $\min{(Q_{+}/k,Q_{-}/k)}\geq c$ can be discussed a bit. In essence, the condition $c^2k=\omega(1)$ (l332) means that the number of clusters should be **big**, especially if a lot of clusters have the same labels (in which case $c\approx 0$). But on the other hand assumption 2.2 (third item) tells that the number of clusters should be small.

Moreover the dimension in which the theorems hold should be high (from what I understand of theorem 3.1), but there shouldn't be too much points per clusters (otherwise $n$ would be too big).

In overall, beyond artificial example 2., is there practical settings that fulfill those hypothesis? I'm scared that the results only apply to a high dimensional setting with very few points per cluster, that are all orthogonal, and that do not overlap.



**Questions:**

## Suggestion

Based on the quality of the literature review, I think a "summary" table that contains a high level description of some significant prior works, with their associated hypothesis and main conclusions, would help to situate the paper better, in the spirit of:

| Paper | Hypothesis on model | Hypothesis on data |  Conclusion
| ----------- | ----------- |  -----------   | -----------
| Paper 1|  two-layers ReLu | none | converges
| Paper 2| homogeneous | almost orthogonal | converges and generalizes
| Our | etc... | clusters  | generalizes but not robust

## Question

How much do the results depend on the hypothesis that the activation is a ReLU ? This is part of the requirements of prior work, and clearly the  case of $\phi(z)\geq z$ (l236). Do you think that extension to similar shaped activations is straighforward, or very difficult?




**Limitations:**

The limitations of the hypothesis are discussed, but this part can be detailed in more depth. I have a few remarks and questions.

l207 : I think the formula can be simplified as

$$k(\max_{i\neq j} |\langle \mu_i,\mu_j\rangle|+C+1)\leq \frac{1}{10}(d - C + 1)$$

where $C$ depends on $d$ and $\sigma$. An interpretation can be given: "The centroids must be almost orthogonal to each other, and there shouldn't be too much of them.". Note sure if $\frac{1}{10}$ plays a special role, or if any other constant small than $1$ could have done the trick in the proofs. It is worth mentioning it, because this condition is hard to interpret.

We see that "it is worth noting that Assumption 2.2 implies that the data is w.h.p. linearly separable" shows that the hypothesis that the clusters are almost orthogonal is quite strong.

What I understand from "Finally, we remark that when k is small, our results may be extended to the case where σ > 1." (l227) hints that what you need is a "almost orthogonal clusters that don't overlap, so they are separable" kind-of hypothesis. I am not sure how much additional insights it brings compared to the work of  Vardi, Yehudai, and Shamir [VYS22].

In particular, the upper bound on $n$ in Thm3.1 is quite surprising, fortunately the discussion l265 "It is noteworthy
that all existing non-vacuous generalization bounds for interpolating nonlinear neural networks in the
presence of label noise require n < d [FCB22; Cao+22; XG23; Fre+23a; Kou+23]." helps to understand why this is required.

---

> ### Author Rebuttal · Authors · 2023-08-08
>
> We thank the reviewer for their detailed review.  We respond to their main questions and comments below.
>
> ### Assumptions on $Q_{\pm}$, $k$ and $c$:
>
>  We apologize for the confusion here.   Although our analysis can accommodate non-constant $c$, for simplicity one can take $c$ to be a constant, in which case the assumption that $\min(Q_+/k, Q_-/k)\geq c$ means that at least a constant fraction of the clusters have positive labels and a constant fraction of the clusters have negative labels. The comment about $c^2 k= \omega(1)$ can then be thought of as a statement about the number of clusters $k$ rather than $c$.  Thus, the lower bound on $k$ is $\omega(1)$, while the upper bound on $k$ from Assumption 2.2 is typically much larger (see the first two items in Example 1). We will be sure to revise our manuscript to make this more clear.
>
>
> ### Assumptions: high-dimensionality, orthogonality etc.
>
> The reviewer is concerned that the results only apply to high-dimensional settings with few points per cluster that are all orthogonal and non-overlapping. We want to emphasize that our assumptions can accommodate low-dimensional settings (i.e., where $n \gg d$) with correlated clusters and many points per cluster. For instance, as in the first bullet in Example 1 and in Example 2, we can permit $k = \tilde{O}(\sqrt{d})$ clusters with cluster correlations of order $\tilde{O}(\sqrt{d})$ and with $\sigma=1$.  Since $\sigma=1$, the cluster radius is roughly $\sqrt{d}$, so that the cluster radius is of the same order as the distance between clusters. And as we mention in lines 253--255 and 328--330, our results hold when $n = \mathrm{poly}(d)$ for any polynomial $\mathrm{poly}()$, so we for instance could have $n = d^{50}$ which is far from a `high-dimensional’ setting. The only setting not covered is when $n$ is super-polynomial in $d$, which we think is a fairly restricted setting but which also would require significant new technical innovations as we mention in line 265. Of course, one can obtain many additional examples such as the above, that satisfy our assumptions.
>
> Regarding practicality of our assumptions, we agree that the setting we consider is somewhat stylized, but we wish to emphasize that in order to characterize whether or not KKT conditions for margin-maximization imply generalization or susceptibility to adversarial attacks for test data, we must make some type of distributional assumption. We think that any distributional assumption comes with benefits and pitfalls, and we are not aware of widely-accepted definitions of ‘real-world distributions’ or ‘practical’ settings. We think the assumptions in our work are fairly natural and uncontrived.  We believe the possibility of a ‘double-edged sword’ of the implicit bias of GF in this setting is a novel and remarkable finding that would be of interest to the NeurIPS community.
>
> ### Suggestion on the ​​literature review:
> We thank the reviewer for the suggestion. In the camera-ready version we will try to situate the paper better in the spirit of the suggested table.
>
> ### Assumption of ReLU activation:
> We think it would be relatively straight-forward to generalize our results to the leaky ReLU, which is also homogeneous and satisfies a similar inequality to the one you state ($\phi(z) \geq \gamma z$ for $\gamma >0$ would suffice in many places), at the expense of additional notation and dependence on the $\gamma$ parameter.  However, we think there would be significant difficulties with extending the result to non-homogeneous settings, since there are essentially no known characterizations of the implicit bias of gradient flow/descent for such cases.
>
>
> ### Additional remarks:
> Thank you for your additional suggestions.  We will elaborate more on Line 207 in the camera-ready as the reviewer suggested.
>
>
> Regarding the comparison with VYS22, as we have mentioned to Reviewer 8zLE, at a high-level the reviewer is correct that there are parallels between clusters in our setting and samples in their setting. However, there are important conceptual differences. First, their results require that the ambient dimension d is much larger than the number of samples n; the high-dimensional setting often has different generalization behavior than in the low-dimensional setting (e.g., overfitting can be 'benign' in the high-dimensional setting, but harmful in low-dimensional setting [1]).  Second, their analysis held for arbitrary labels $y_i$ of the training data, and it was unclear how an underlying distribution over x and a signal (in the form of a conditional distribution of $y|x$) would affect both generalization and the existence of adversarial test examples. We will be sure to add details in the revision which make this more explicit.

---

> > ### Comment · Reviewer_ZRWm · 2023-08-14
> >
> > Thank you for your detailed answer and for your clarifications.
> >
> > >  so we for instance could have $n=d^{50}$ which is far from a `high-dimensional’ setting.
> >
> > While I agree with your overall answer, I believe that as long as the number of points is polynomial in the dimension, we are *asymptotically* in a high dimensional setting. Here, for $d\gg 50$, $n=d^{50}$ is far lower than the number of corners $2^ d$ of the hyper-cube.
> >
> > > Assumption of ReLU activation:
> >
> > Thank you for your answer. I have an additionnal question: do you think it extends to the case of non elementwise activation functions that are homogeneous? I am thinking of MaxMin for example, that operates on pairs of consecutive neurons.
> >
> > $$\text{MaxMin}([x,y])=[\min(x,y),\max(x,y)].$$

---

> > > ### Author Response · Authors · 2023-08-14
> > >
> > > > While I agree with your overall answer, I believe that as long as the number of points is polynomial in the dimension, we are asymptotically in a high dimensional setting. Here, for $d\gg 50$, $n=d^{50}$ is far lower than the number of corners of the hyper-cube.
> > >
> > > We are struggling to understand the reviewer’s definition of 'high-dimensional'.  Our understanding of 'high-dimensional’ refers to settings where the number of samples is either of the same order or much larger than the number of samples; this is the way 'high-dimensional’ is used in Wainwright’s textbook on *High-Dimensional Statistics*, for example.  With this definition, in the high-dimensional setting samples are nearly-orthogonal and there are few samples per cluster, which are some of the phenomena the reviewer expressed concern about in their review.  But these phenomena do not hold when $n$ is a large polynomial in $d$ as our settings allow.
> > >
> > > We are also confused by what the reviewer means by 'asymptotic’ as our results can be applied for finite $n$ and finite $d$.
> > >
> > > > Thank you for your answer. I have an additionnal question: do you think it extends to the case of non elementwise activation functions that are homogeneous? I am thinking of MaxMin for example, that operates on pairs of consecutive neurons.
> > >
> > > We think extending our analysis to non-elementwise homogeneous activations like maxpooling is an intriguing direction for future research.  We are not sure whether our results would hold for the MaxMin activation.

---

> > > > ### Comment · Reviewer_ZRWm · 2023-08-14
> > > >
> > > > >  Our understanding of 'high-dimensional’ refers to settings where the number of samples is either of the same order or much larger than the number of samples; this is the way 'high-dimensional’ is used in Wainwright’s textbook on High-Dimensional Statistics, for example. With this definition, in the high-dimensional setting samples are nearly-orthogonal and there are few samples per cluster, which are some of the phenomena the reviewer expressed concern about in their review. But these phenomena do not hold when is a large polynomial in as our settings allow.
> > > >
> > > > Thank you for your clarification.
> > > >
> > > > By high dimensional I informally meant settings in which it is hard to "cover" the whole space with samples (if we think about covering numbers, for example). For example, when $n = \text{poly}(d)$, for $d$ *high enough* (this is what I meant by *asymptotic*), when are points are sampled on the corners of the hyper-cube independantly and uniformly, that all end-up at approximately the same distance from each other (even though $n\gg d$) which is also a typical behavior of the high dimension. Therefore, for me, the case $n = \text{poly}(d)$ when $n,d\rightarrow \infty$ also deserve the "high dimension" name. The exponential dependency of the number of samples in the dimension is typical from results related to the "Curse of dimensionality".
> > > >
> > > > > We are also confused by what the reviewer means by 'asymptotic’ as our results can be applied for finite and finite .
> > > >
> > > > It was not a critic directed towards your work, your definition nor your rebuttal, it was rather a general discussion about what should be considered "high dimensional". I ackowledged that your results are valid for finite $n$ and $d$.

---

> > > > > ### Author Response · Authors · 2023-08-14
> > > > >
> > > > > > By high dimensional I informally meant settings in which it is hard to "cover" the whole space with samples (if we think about covering numbers, for example). For example, when , for high enough (this is what I meant by asymptotic), when are points are sampled on the corners of the hyper-cube independantly and uniformly, that all end-up at approximately the same distance from each other (even though ) which is also a typical behavior of the high dimension. Therefore, for me, the case when also deserve the "high dimension" name. The exponential dependency of the number of samples in the dimension is typical from results related to the "Curse of dimensionality".
> > > > >
> > > > > Thank you for the clarification. If we understand correctly, then by the reviewer’s definition of high dimensionality, every polynomial-time learning algorithm (i.e., runs in $\mathrm{poly}(d)$ time with $\mathrm{poly}(d)$ sample complexity) operates in a high dimensional regime.
> > > > >
> > > > > Regardless of how we define "high dimensional", the important thing is that we all agree that our results hold if $n=\mathrm{poly}(d)$ and do not hold $n=\exp(d)$.

---

### Official Review · Reviewer_kUQf · 2023-07-06

**Soundness:** 3 good
**Presentation:** 3 good
**Contribution:** 3 good
**Rating:** 5
**Confidence:** 1

**Summary:**

The paper studies the implicit regularization brought by the neural network itself. The authors theoretically prove that in two-layer ReLU networks trained with the logistic loss or the exponential loss, the implicit bias would lead to the solutions that generalize well but non-robust, regardless of the size of the network.

**Strengths:**

- The topic is valuable and may shed some light on the field.
- The theoretical proof seems to be sound.

**Weaknesses:**

- Lacking of even toy experiments.

**Questions:**

N/A

---

### Official Review · Reviewer_8zLE · 2023-07-06

**Soundness:** 4 excellent
**Presentation:** 4 excellent
**Contribution:** 3 good
**Rating:** 7
**Confidence:** 3

**Summary:**

The authors show under a special data cluster model, the implicit bias of gradient flow converge to KKT points that generalize well but are not robust under l2 perturbations, when robust solutions of the problem exists. Their results are built upon earlier works on KKT points by LL20 and JT20, and the more recent work by VSS22, and extend these results further.


**Strengths:**

- The authors study the gradient flow solution of 2-layer ReLU network on a mixture of Gaussian distribution, and prove that while the solution generalizes well (classifies almost every test point correctly with high probability), it is not robust (can change its classification with perturbation much less than \sqrt{d}, the optimal achievable robustness). This is a very nice result following up on the previous line of work on this topic.

- The paper is well-written and clear. While the result is technical the presentation is easy to follow and main proof ideas succinctly presented in the limited space.


**Weaknesses:**

- Given the assumption on large cluster separation (\sqrt{d} between cluster centers) and small variance of the clusters (\sigma<1), it feels like the clusters behave very much like individual isolated points/samples. In view of this, the differentiation with the results of VSS22 in the introduction section appear to be weaker than the authors claim, as the cluster means take the role of training samples in VSS22, which still requires near-orthogonality. Also, results wrt test data is also easier under the data assumptions used in this paper because test data essentially takes the same label from their respective cluster means.


**Questions:**

- Although not strictly necessary for a theroy paper, it would be nice to have some simulations on synthetic data since it is easy to set up with the data model the authors assume.



**Limitations:**

- The authors state clearly the assumptions used for their results. Negative societal impacts not applicable here.

---

> ### Author Rebuttal · Authors · 2023-08-08
>
> We thank the reviewer for their positive review of our work.  We respond to two comments/questions below:
>
> ### Difference with VYS22:
> At a high-level, the reviewer is correct that there are parallels between clusters in our setting and samples in their setting. However, there are a few important differences.  The first is that when $\sigma=1$ the distance between points in the same cluster is of the same order as the distance between points in different clusters (or the distance between cluster means). So the clustered setting is significantly different from the isolated points from VYS22.  The second is a more conceptual difference, which is that their results require that the ambient dimension $d$ is much larger than the number of samples $n$; the high-dimensional setting often has different generalization behavior than in the low-dimensional setting (e.g., overfitting can be 'benign' in the high-dimensional setting, but harmful in low-dimensional setting [1]).  Moreover, their analysis held for arbitrary labels $y_i$ of the training data, and it was unclear how an underlying distribution over $x$ and a signal (in the form of a conditional distribution of $y|x$) would affect both generalization and the existence of adversarial test examples.
>
>
>
> ### Experiments:
> As we mention to Reviewer DAg1, although we agree that a thorough experimental investigation of our results would be beneficial, we are not convinced the time and paper-space needed to do them well would be worthwhile.  In particular, although it would be easy to verify the generalization properties and that the universal perturbation $z$ we discover indeed succeeds in adversarial attacks, we believe that simply verifying our theorems are correct with experiments would add little value. However, we agree that a thorough empirical study that extends our setting and evaluates whether the "double-edged sword" effect of the implicit bias occurs more generally is an intriguing topic for future research.
>
>
> [1] Guy Kornowski, Gilad Yehudai, Ohad Shamir. From Tempered to Benign Overfitting in ReLU Neural Networks. arXiv preprint 2305.15141

---

> > ### Comment · Reviewer_8zLE · 2023-08-17
> >
> > I would like to thank the authors for clarifying the differences of the current work with VYS22.

---

### Official Review · Reviewer_DAg1 · 2023-07-07

**Soundness:** 4 excellent
**Presentation:** 4 excellent
**Contribution:** 3 good
**Rating:** 7
**Confidence:** 4

**Summary:**

This paper studies the generalization and robustness of solutions obtained by gradient flow on two-layer ReLU networks. Under a distributional setting where the data is sampled from a Gaussian mixture distribution, this paper shows that the gradient flow is biased towards solutions that generalize well, but are vulnerable to adversarial examples. The theorems are built on LL20 and JT20 (Theorem 2.1). The authors also show the existence of a robust solution in this setting and prove that any solution obtained via gradient flow is non-robust.  Although the assumption is a bit restrictive, the result is novel and interesting. The authors also provide a few examples which help to understand the assumptions.

**Strengths:**

1. This paper is well-written and easy to follow. The problem is well-motivated. The authors provide a thorough discussion of the related work.
2. The theoretical results are solid and interesting, the proof ideas help to understand the results.
3. This is an active area of research. The contribution is relevant.

**Weaknesses:**

1. The authors assume that when the training loss is small, the gradient flow will start to converge to a KKT point of some maximum margin problem. What if we have a random initialization, how is this small training loss guaranteed?
2. This might be relatively minor, but numerical experiments would help to demonstrate the vulnerability (non-robustness) of the training of two-layer ReLU networks.

**Questions:**

1. The authors emphasize that the result hold in the rich regime. Could the authors be more specific, as there is no discussion about the rich regime in the results?
2. Does such a result hold for regression, in other words, what about other loss functions, such as squared error?

**Limitations:**

The authors assume that the data is clustered and the number of clusters is not too large, therefore the data is linearly separable. This is a relatively strong assumption. The relaxation of this assumption and extensions to other settings would be interesting.

---

> ### Author Rebuttal · Authors · 2023-08-08
>
> We thank the reviewer for their comments and suggestions.  We respond to specific points below.
>
> ### Assumption of small training loss:
>
> Indeed, the implicit bias result kicks in when gradient flow reaches a sufficiently small training loss. One of the advantages of relying on the KKT conditions of the max-margin problem instead of analyzing the full trajectory of gradient flow is that it allows us to separate the convergence question from the generalization and robustness questions. Hence, even in settings where proving convergence is difficult, we may prove generalization and non-robustness.  Nevertheless, as the reviewer mentioned, proving convergence in our setting is an interesting question. For wide networks convergence can be shown by an NTK analysis, and in the general case this question is open. We will add a discussion on this issue.
>
>
> ### Experiments:
> Although we agree that a thorough experimental investigation of our results would be beneficial, we are not convinced the time and paper-space needed to do them well would be worthwhile.  In particular, although it would be easy to verify the generalization properties and that the universal perturbation $z$ we discover indeed succeeds in adversarial attacks, we believe that simply verifying our theorems are correct with experiments would add little value.
>
> More broadly, we think it is worth emphasizing that this work is a part of a series of works on understanding the theoretical foundations of robustness in deep learning.  Many of these works do not include experiments and have been published at NeurIPS, including Bubeck and Sellke’s Outstanding Paper Award-winning work at NeurIPS 2021.  However, we agree that a thorough empirical study that extends our setting and evaluates whether the "double-edged sword" effect of the implicit bias occurs more generally is an intriguing topic for future research.
>
>
> ### Rich regime:
> Here we are re-using the terminology from, e.g., [1], where by "rich regime" refers to neural network training which does not lie in the "kernel regime".  The "kernel regime" requires a number of assumptions regarding the network width, initialization, etc, and since our setting makes no such assumptions our results hold for networks in the "rich regime".  We shall clarify this in the camera ready.
>
> ### Other losses:
> We are quite interested in exploring the questions in this paper in the regression setting.  However, one difficulty is that there is a much less developed theory about the implicit bias of gradient flow/descent in neural networks with regression losses, which is provably not the minimum-l2 loss in relu networks [2].  This makes it difficult to use our approach which relies upon rather explicit characterizations of the implicit bias of optimization algorithms to characterize the behavior of trained neural nets.
>
>
> [1] Blake Woodworth, Suriya Gunasekar, Jason D. Lee, Edward Moroshko, Pedro Savarese, Itay Golan, Daniel Soudry, Nathan Srebro. Kernel and Rich Regimes in Overparametrized Models. COLT 2020
>
> [2] Gal Vardi and Ohad Shamir. Implicit Regularization in ReLU Networks with the Square Loss. COLT 2021

---

> > ### Comment · Reviewer_DAg1 · 2023-08-17
> >
> > Thank the authors for detailed explanation. I would like to increase my rating from 6 to 7.

---

### Decision · Program_Chairs · 2023-09-21

**Decision:**

Accept (poster)

**Comment:**

The paper studies how gradient descent in over-parameterized ReLU networks leads to solutions that generalize well, but lack robustness. Reviewers found the theoretical results interesting and useful, and the paper well written.